# Multiple drivers and lineage-specific insect extinctions during the Permo–Triassic

Corentin Jouault [1,2,3] ✉, André Nel[1], Vincent Perrichot [2],
Frédéric Legendre [1,4] & Fabien L. Condamine [3,4] ✉

The Permo–Triassic interval encompasses three extinction events including the most dramatic biological crisis of the Phanerozoic, the latest Permian mass extinction. However, their drivers and outcomes are poorly quantified and understood for terrestrial invertebrates, which we assess here for insects. We find a pattern with three extinctions: the Roadian/Wordian (≈266.9 Ma; extinction of 64.5% insect genera), the Permian/Triassic (≈252 Ma; extinction of 82.6% insect genera), and the Ladinian/Carnian boundaries (≈237 Ma; extinction of 74.8% insect genera). We also unveil a heterogeneous effect of these extinction events across the major insect clades. Because extinction events have impacted Permo–Triassic ecosystems, we investigate the influence of abiotic and biotic factors on insect diversification dynamics and find that changes in floral assemblages are likely the strongest drivers of insects' responses throughout the Permo–Triassic. We also assess the effect of diversity dependence between three insect guilds; an effect ubiquitously found in current ecosystems. We find that herbivores held a central position in the Permo–Triassic interaction network. Our study reveals high levels of insect extinction that profoundly shaped the evolutionary history of the most diverse non-microbial lineage.

Over geological times, the evolution of marine and terrestrial organisms has been punctuated by major diversification and extinction events[1–3]. The most dramatic of these extinctions occurred at the boundary of the Permian and Triassic periods, ≈252 million years ago (Ma), and is known as the latest Permian mass extinction (LPME)[4,5]. Famously renowned as the Phanerozoic's most severe extinction, the LPME is commonly named the 'Great Dying' because more than 90% of the biodiversity (taxonomic richness) is supposed to have disappeared[6–8]. Two additional extinctions are sometimes recorded in the vicinity of the LPME: the Guadalupian extinction event (GEE), which occurred ≈260.5 Ma[9]; and the Carnian pluvial episode (CPE) that occurred between 234 to 232 Ma[10–12]. The CPE is thought to have been caused by a combination of volcanic emissions, subsequent warming, and the dissociation of methane clathrates[11]. It is marked by global warming and four episodes of increased rainfall leading to global environmental changes (e.g. modification of floral assemblages and plant cover[12,13]). The GEE is thought to be linked to the equatorial eruption of the Emeishan Traps, which caused a cascading effect with sudden cooling followed by global warming while also affecting ocean acidity[14–16]. Similarly, eruptions of the Siberian Traps Large Igneous Province and global warming are considered major triggers of the LPME[17,18] resulting in severe environmental disruptions. However, the impact of these crises has mainly been quantified for marine and vertebrate biodiversity[2,19]. It has even been challenged for land plants[20], questioning the coeval and synchronicity of declines in terrestrial invertebrates, particularly in insects[7,21–23].

Because extant diversity stems from the outcomes of these events (selections and demises of lineages), it is crucial to assess how they

[1]Institut de Systématique, Évolution, Biodiversité (ISYEB), Muséum national d'Histoire naturelle, CNRS, Sorbonne Université, EPHE, Université des Antilles, CP50, 57 rue Cuvier, 75005 Paris, France. [2]Univ. Rennes, CNRS, Géosciences Rennes, UMR 6118, F-35000 Rennes, France. [3]CNRS, UMR 5554 Institut des Sciences de l'Évolution de Montpellier, Place Eugène Bataillon, 34095 Montpellier, France. [4]These authors jointly supervised this work: Frédéric Legendre, Fabien L. Condamine. ✉e-mail: jouaultc0@gmail.com; fabien.condamine@gmail.com

operated and shaped insect diversity in the deep time. Previous family-level studies repeatedly suggested that the LPME is the most severe insect extinction, but also raised doubts about the impact of the GEE or CPE on the evolutionary dynamic of insect diversification[7,21,23]. Importantly, many extant insect orders originated and/or diversified during the Permian and the Triassic[24,25], further challenging the idea of major extinctions during these periods, and instead supporting diversification episodes. As for the GEE and CPE events, few analyses investigated their effect at the genus level. No consensus has been reached on their possible impact on insects.

The deep-time dynamic of insect diversification (i.e. variations in origination and extinction rates) has supposedly been affected by abiotic factors such as global temperature or continental fragmentation[7,23,26–30]. However, their role throughout the Permo–Triassic crises, as well as that of biotic factors such as interactions (diversity dependence) between ecological groups, have never been investigated. We defined two categories of diversity dependence with the effect of a clade's diversity changes: (i) on its own speciation and extinction rates (i.e. intra-clade diversity dependence) or (ii) on the speciation and extinction rates of another co-occurring clade (i.e. inter-clade diversity dependence). Both intra-clade and inter-clade interactions may occur for taxa with similar ecology or under limited resource conditions[26,31,32]. Accordingly, a lineage can thrive, decline, or even be replaced by another following the freeing or filling of an ecological niche[32–34]. For insects, whether the decline of herbivores has had a cascading effect on other ecological guilds (e.g. predators) remains to be fully deciphered, a significant challenge because the diet and ecology of fossil insects remain poorly known[35].

Here, using the insect fossil record, we aim to assess the impact of the Permo–Triassic crises on insects and decipher the likely triggers of their diversification dynamics. This fossil record is relatively well-documented over the Permian and Triassic periods, without major gaps[23]. Over an Asselian to Rhaetian timespan (i.e. between 298.9 and 201.3 Ma), we compiled and analysed a new dataset of over 17,250 worldwide insect occurrences, vetted at the species level for 1784 genera and at the stage- or formation-level for ages. Relying on a Bayesian framework that estimates process-based birth–death models while incorporating the preservation process and uncertainties associated with fossil occurrences[36,37], we investigated whether and to what extent origination (speciation at the genus level) and extinction rates responded to environmental changes throughout the period encompassing putative sudden and strong extinction events (i.e. CPE, LPME and GEE). We simultaneously assessed the effect of diversity changes between three or four guilds (herbivores, predators, detritivores/fungivores, and generalists; the last two were merged in a guild dubbed 'others' in the three-guilds model) on their origination and extinction rates by quantitatively investigating the roles of diversity dependence among insect clades throughout the Permian and Triassic periods.

## Results and discussion

### Unexpected and heterogenous extinctions during the Permian and Triassic

Our fossil dataset included 14,483 occurrences assigned at the family level representing 418 families and 14,789 occurrences assigned at the genus level representing 1784 genera (Supplementary Data 1: Genus_level). We inferred the diversification history of insects relying on both the birth–death model with constrained shifts (BDCS[38]) and the reversible-jump Markov Chain Monte Carlo model (RJMCMC[37]), both at genus and family levels. Previous studies have suggested important extinctions of insect families during the Guadalupian or the LPME but have never investigated the decline of insects at the genus level[7,21,23,39].

The genus-level analyses revealed that the diversity dynamic of insect genera declined three times between the Guadalupian and the Late Triassic (Fig. 1a–c and Supplementary Figs. 1–11), with large decreases in net diversification rates (Fig. 1b; defined as origination

minus extinction), while family-level analyses indicated a decline in insect families' diversification around the Ladinian/Carnian (L/C) boundary (Fig. 1d–f and Supplementary Figs. 12–16). We ensured that singletons—a classical sampling bias—did not alter the inferred patterns and found that only the magnitude of the shifts was modified (Supplementary Figs. 6, 16). These changes in genus-level diversity dynamic are characterised by peaks of extinction rates at the Roadian/Wordian (R/W; ≈1.8-fold background extinction rate) and the Ladinian/Carnian (≈3.9-fold background extinction rate) boundaries (266.9 Ma and 237 Ma, respectively), but most notably during the LPME (≈4.3-fold background extinction rate), indicating that insects suffered a loss of generic diversity during these events (Fig. 1b and Supplementary Figs. 1–11). The LPME impacted most major insect lineages (Polyneoptera, ≈4.9-fold background extinction rate, Fig. 2a–c; Holometabola, ≈3.3-fold background extinction rate, Fig. 2d–f; Acercaria, ≈6.1-fold background extinction rate, Fig. 2g–i), a result corroborating previous studies on insect diversification[7,21,23,40]. The R/W extinction impacted mostly the Polyneoptera (≈1.7-fold background extinction rate, Fig. 2a–c) with an important decline of Protelytroptera and several lineages of Archaeorthoptera (Supplementary Fig. 9). The L/C impact is recorded across several insect lineages (Fig. 2 and Supplementary Figs. 1–11) but mainly for the Holometabola (≈6.9-fold background extinction rate, Fig. 2d–f) and Polyneoptera (≈5.1-fold background extinction rate, Fig. 2a–c). Quantifying the effect of each extinction event on insect genera, we found that the LPME had the strongest impact with the demise of 82.6% of genera, followed by the L/C impact with 74.8%, while the effect of the R/W extinction was less pronounced with the extinction of 64.5% of insect genera (Supplementary Table 1). Importantly, it is the first time, to our knowledge, that extinction peaks are recorded at the R/W and L/C boundaries for insects, while a huge loss of diversity is known for vertebrate clades at this time[41,42]. Conversely, the GEE and CPE are not recorded, and the R/W or L/C extinctions cannot be mistaken for these events because they occurred long before them.

At the family level, no evidence of the three extinction events was found, although the L/C boundary showed a period of insect extinction (≈4.7-fold background extinction rate; Fig. 1d–f). The net diversification of Acercaria and Holometabola, stable since the Guadalupian, decreased around the L/C boundary but without a simultaneous increase in extinction rate (Supplementary Figs. 12, 13). In contrast, the net diversifications of the Palaeoptera and Polyneoptera were stable throughout the entire time interval (Supplementary Figs. 14, 15). Our results strikingly differ from all previous analyses by suggesting that extinctions occurred, at the family level, around the L/C boundary and that concerned families were not assigned to one of the four major clades of insects[7,21,23]. The largest extinctions of insect families may have rather occurred at the end of the Triassic period, with the demise of insect families belonging to ancient orders (i.e. that began to radiate during the Carboniferous and/or Permian like Caloneurodea, Permopsocida, Mecoptera and Plecoptera), as evidenced by species of these groups recorded during the Triassic but not thereafter[7,26,43].

How lineages cope with and recover from extinction is an important concern in organism evolutionary dynamics. Here, insect genera diversified (origination exceeding extinction) just before the LPME (≈3.5-fold background origination rate) and during the Middle Triassic (≈3.8-fold background origination rate), suggesting they had recovered from previous crises (Fig. 1a, d). In other words, when a portion of insect diversity went through an extinction crisis at this time, it quickly returned to a pre-crisis diversity level, a phenomenon also recorded in ammonoids and called flash recovery[44]. The LPME was characterised by overlapping high origination (≈0.45 events/Ma/lineage) and extinction (≈0.57 events/Ma/lineage) rates (Fig. 1a). This extinction is well established and directly evidenced in the fossil record. It corresponds to the transition from the Palaeozoic Insect Fauna (e.g. extinction of Palaeodictyopteroidea; Supplementary

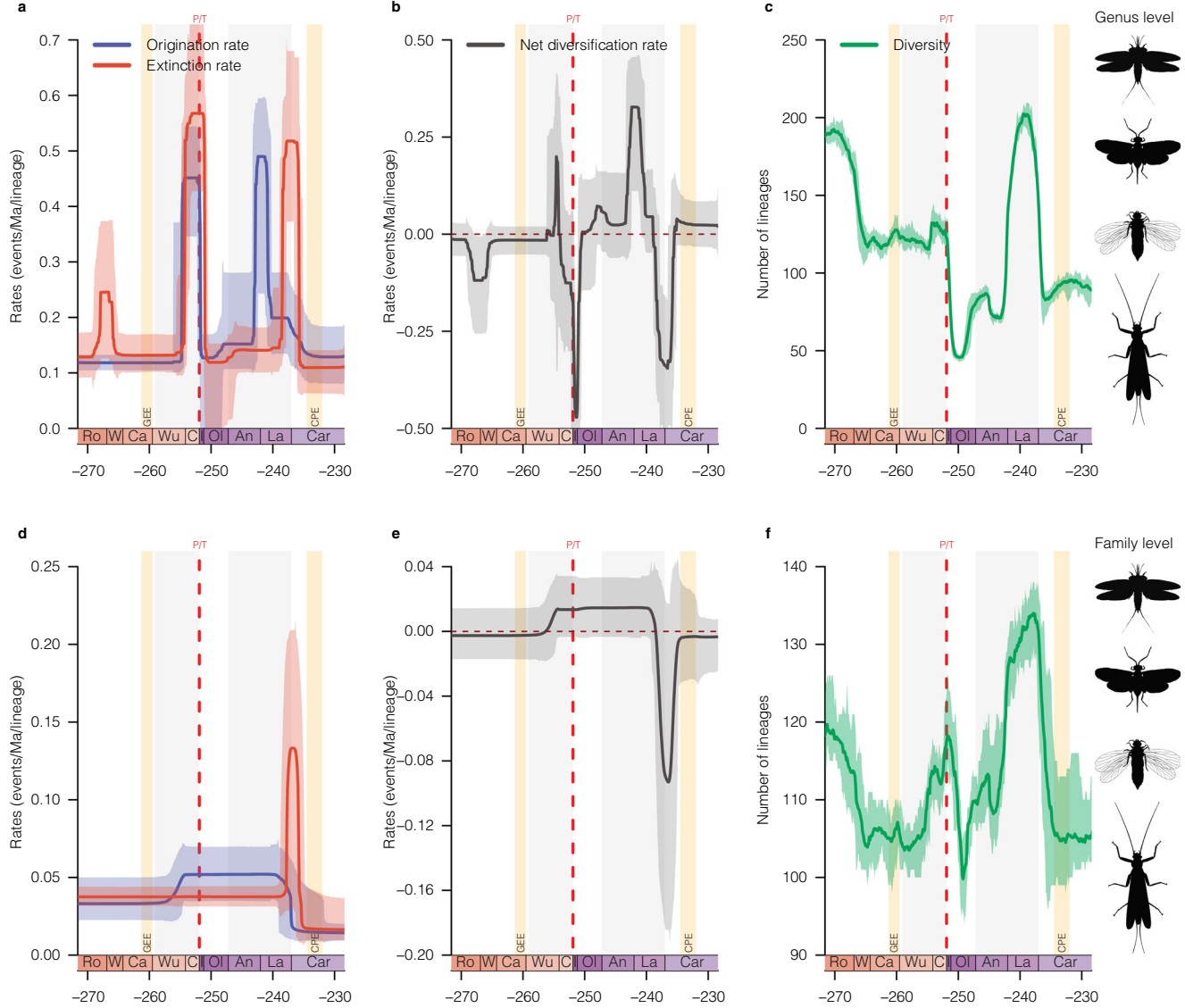

**Fig. 1 | Diversification and diversity dynamics of insects during the Permo–Triassic.** Bayesian estimates of origination (blue), extinction (red), net diversification (black, origination minus extinction) rates, and diversity of all insects at the genus level (**a**–**c**) and family level (**d**–**f**). Solid lines indicate mean posterior rates and shaded areas show 95% CI. Net diversification decreased three times; around Roadian/Wordian (R/W), Permian–Triassic (P/T), and Ladinian/Carnian (L/C) boundaries. The diversities of all insect genera and families were in decline just before the LPME and the L/C boundaries. Reconstructions of diversity trajectories replicated to incorporate uncertainties around the ages of the fossil

occurrences. The colour of each period in the chronostratigraphic scale follows that of the International Chronostratigraphic Chart (v2022/02). Ro Roadian, W Wordian, Ca Capitanian, Wu Wuchiapingian, C Changhsingian, I Induan, Ol Olenekian, An Anisian, La Ladinian, Car Carnian, GEE Guadalupian extinction event, P/T Permo–Triassic mass extinction (251.902 Ma), and CPE Carnian pluvial episode. Insect silhouettes from http://phylopic.org/. Hymenoptera, Miomoptera, Palaeodictyoptera and Plecoptera licences at https://creativecommons.org/publicdomain/zero/1.0/.

Fig. 11) to the Mesozoic Insect Fauna (e.g. first representatives of the crown Diptera and Hymenoptera, the rise of Hemipteroidea and diversification of Coleoptera)[45,46].

Dramatic events such as the R/W, the LPME and the L/C are often investigated in large-scale analyses, suggesting that they might have affected all insect lineages uniformly[7,21,23]. In contrast, we found a heterogeneous impact of all these events on insect clades (with no extinction for the Palaeoptera, Fig. 2j–l), and at different taxonomic levels (Figs. 1, 2 and Supplementary Figs. 1–16). Similarly, we found no extinction at the family level for all the major clades of insects (Supplementary Figs. 12–16). We assume that the differences between our results and the global patterns previously inferred come from the recent and significant increase in the knowledge of fossil insects from the Permian and Triassic periods[25,40,47]. The extensions of numerous family life spans and the description of new

representatives after the Guadalupian and LPME events are indicative of the resilience of insects to these events[48]. Additionally, the taxonomic level to which some of our analyses are conducted (genus level) greatly differs from family-level data used in previous studies and may explain the differences we recorded. These results point to the importance of up-to-date data and the need to better document the fossil record to gain knowledge on the 'life span' of lineages. Similarly, they emphasise the importance of investigating extinction events at different taxonomic levels and interpreting general patterns with caution.

**Extinction increases with taxon age during the Permo–Triassic interval**

Because we found that the LPME, the R/W and the L/C had heterogeneous impacts of different durations and magnitudes, we estimated

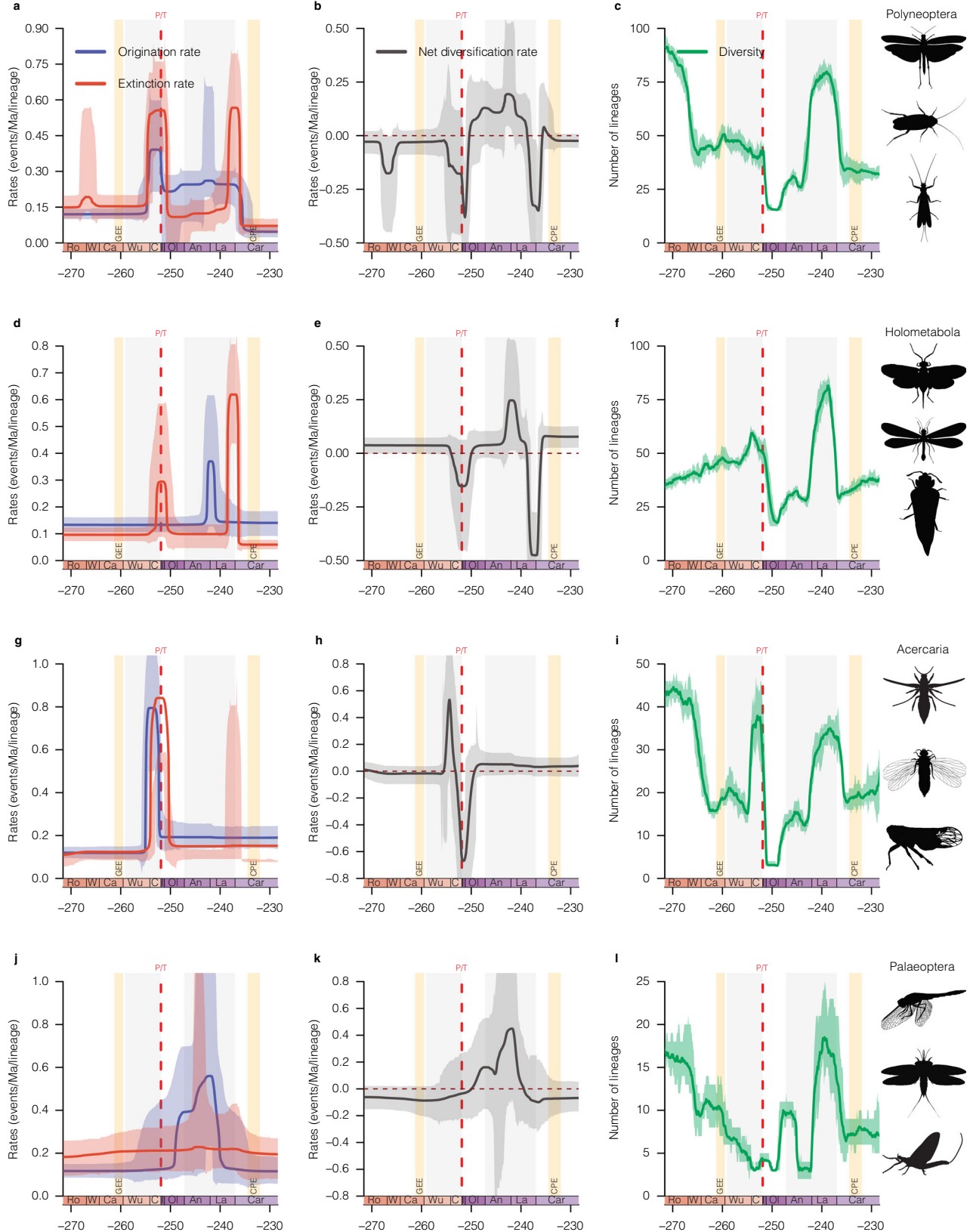

whether the extinction was dependent on taxon age[49] using the age-dependent extinction model (ADE[50]). We investigated the ADE effect in periods of stable net diversification, i.e. before the LPME (264.28 to 255 Ma; hereafter pre-decline period) and after the CPE (234 to 212 Ma; hereafter post-crisis period). We also tested the ADE model during the decline of the LPME (from 254.5 to 251.5 Ma).

No effect of age on extinction rates was found during the pre-decline and post-crisis periods (respectively $\phi = 2.258$ and 1.766, 95% credibility interval [CI] = 0.61–4.725 and 0.7421–3.2443; Supplementary Table 2), which supports Van Valen's hypothesis of constant extinction. However, during the LPME, age had a strong effect on extinction ($\phi = 9.1677$, 95% CI = 2.3886–18.7343; Supplementary

**Fig. 2 | Diversification and diversity dynamics of major clades of insects, inferred from genus level analyses, during Permo–Triassic. a, d, g, j** Bayesian estimates of origination (blue) and extinction (red) rates for each of four major clades of insects. **b, e, h, k** Net diversification rate (origination minus extinction) of four major clades of insects showing heterogeneous response to the three extinctions event (R/W; LPME or P/T; and L/C). **c, f, i, l** Diversity (number of genera) of four major clades of insects shows heterogeneous variation throughout Permo–Triassic. For each plot, solid lines indicate mean posterior rates; shaded areas show 95% CI. Reconstructions of diversity trajectories replicated to incorporate uncertainties around the ages of the fossil occurrences. Colour and abbreviations as in Fig. 1. Insect silhouettes from http://phylopic.org/. Caelifera (by Didier Descouens; vectorized by T. Michael Keesey) licences at https://creativecommons.org/licences/by-sa/3.0/; Coleoptera, Dictyoptera, Hemiptera, Hymenoptera, Odonatoptera, Mecoptera, Miomoptera, Palaeodictyoptera and Plecoptera licences at https://creativecommons.org/publicdomain/zero/1.0/; Ephemeroptera and Thysanoptera by Corentin Jouault.

Table 2). This indicates that extinction probability increases with lineage age in an extinction context and shows that Van Valen's law may only apply for stable periods. A similar outcome was found for dinosaurs[51], whereas studies on planktonic foraminifera and Carnivora found that the recently originated clades were more likely to become extinct than older ones[50,52,53]. We speculate that old insect lineages were adapted to pre-crisis environments and could not adapt rapidly enough to their changing environment, e.g. the transition from glossopterid forests to forests composed of pteridophytes and small-leafed sclerophyllous gymnosperms during and after the Permo–Triassic boundary[54].

## Potential ecological drivers of insect diversification and decline

Several ecological factors may have triggered the periods of decline and diversification inferred. Abiotic and biotic factors could explain these dynamics, but to date, no study has investigated the correlations between the extinction and origination of insects and these environmental changes. Using a multivariate birth–death (MBD) model[55], we analysed simultaneously the effect of nine potential drivers (see Selection of abiotic and biotic variables) on insect diversification patterns during the Permo–Triassic. To estimate the role of diversity dependence, we also included the past diversity fluctuations of all insects as an additional variable in the MBD model; we discuss its effect below (Diversity dependence within and between insect clades).

Among all nine variables tested, the MBD model indicated that several factors are correlated with the diversification of insects during the Permo–Triassic, and at different time intervals (Fig. 3). Interestingly, variables related to plant diversity were nearly always found as significant potential drivers of insect evolution (Fig. 3a, b, h, i, k, l). During the Permo–Triassic interval (Fig. 3a, b), the relative diversity of gymnosperms, Polypodiales ferns, and non-Polypodiales ferns is strongly correlated with insect diversification ($\omega > 0.90$), suggesting they were potentially the main drivers of insect diversity dynamics. Origination correlated negatively with the relative diversity of Polypodiales ferns ($G\lambda = -30.83$, 95% CI = $-44.87–-19.52$) and positively with the relative diversity of non-Polypodiales ferns ($G\lambda = 3.42$, 95% CI = 0.97–5.10; Supplementary Table 3). Modifications of floral assemblages have likely impacted the origination of insects, with increasing relative diversity of Polypodiales ferns slowing down insect diversification, while the fluctuation in relative diversity of non-Polypodiales ferns would have accelerated their diversification (Fig. 3a, b). Extinction correlated negatively with the relative diversity of gymnosperms ($G\mu = -5.02$, 95% CI = $-8.50–-1.52$) and positively with the relative diversity of non-Polypodiales ferns ($G\mu = 2.71$, 95% CI = 1.15–4.49). Thus, an increase in the diversity of gymnosperms would have slowed down insect extinction, while an increase in the diversity of non-Polypodiales ferns would have accelerated their extinction (Fig. 3a, b).

Applying the MBD model to the Permian period (Fig. 3i), the only abiotic factor at play was the global variation of atmospheric $O_2$ (Supplementary Table 7), which significantly and negatively correlated with insect origination ($G\lambda = -4.29$, 95% CI = $-6.75–-2.05$), which indicates that the decrease in atmospheric $O_2$ may have slowed down insect diversification. During the Triassic (Fig. 3k, l), all but one (atmospheric $CO_2$) of the tested factors may have driven the diversification dynamics of insects (Supplementary Table 8). Origination was negatively correlated with the variation of atmospheric $O_2$ ($G\lambda = -7.29$, 95% CI = $-2.95–-1.43$)—a similar correlation was found for the Permian (Fig. 3i)—and the diversification of Polypodiales ferns ($G\lambda = -25.24$, 95% CI = $-38.30–-10.64$), suggesting that the decrease in atmospheric $O_2$ and the diversification of Polypodiales ferns have slowed down insect diversification. Origination was also positively correlated with the fluctuation in non-Polypodiales ferns abundance ($G\lambda = 5.64$, 95% CI = 3.56–7.60), suggesting that those plants have facilitated the diversification of insects (Fig. 3k). On the other hand, extinction was negatively correlated with the fragmentation of continents ($G\mu = -75.35$, 95% CI = $-131.20–-35.56$), the diversification of gymnosperms ($G\mu = -13.30$, 95% CI = $-16.78–-9.86$), and the decrease of spore-plants ($G\mu = -27.98$, 95% CI = $-37.13–-18.18$; Fig. 3l). These results indicate that insect extinction decreased when continents get fragmented, and when the diversity of gymnosperms and spore-plants increased. Extinction positively correlated with the diversification of Polypodiales ferns ($G\mu = 40.48$, 95% CI = 21.41–57.11), the decrease of global temperature ($G\mu = 0.15$, 95% CI = 0.04–0.27), and the fluctuation in non-Polypodiales ferns diversity ($G\mu = 7.61$, 95% CI = 5.49–9.72; Fig. 3l). Therefore, the diversification of Polypodiales ferns and non-Polypodiales ferns, as well as the drop in global temperature, would have accelerated insect extinction in the Triassic.

From the damages they left on plants, it appears clearly that herbivorous insects mostly fed on gymnosperms, non-Polypodiales ferns during the Permo–Triassic[56,57], *plus*, Polypodiales ferns in the Triassic. The Polypodiales ferns appeared at the end of the Permian or during the early Triassic and experienced a burst of diversification later during the Mesozoic, while the non-Polypodiales ferns are ancient lineages that declined abruptly during the Permo–Triassic[55]. It is thus unsurprising that insects significantly suffered from the wax and wane of these plant groups, in particular ferns and gymnosperms. This result coincides with the replacement and changes in Permian forests, from dominant glossopterids to pteridophytes and small-leafed sclerophyllous gymnosperms, during and after the Permo–Triassic boundary[54,58,59]. Changes in floral assemblages have likely constrained an ecological shift in insect guilds, strongly contributing to the demise of the Palaeozoic insect fauna and the rise of the Mesozoic insect fauna, because insects already maintained close interactions with plants during the Permian and Triassic periods[57,60–62]. These changes would have favoured insects able to feed on the new dominant plants or to adapt to novel environmental constraints (e.g. Hymenoptera and Coleoptera), while the others collapsed (e.g. Palaeodictyopteroidea). The amber production of the Carnian might also reflect the beginning of resin production as a defence strategy against phytophagous insects, maybe following the rise of Mesozoic insects[63].

The decrease in global temperature and changes in the atmospheric concentration of $O_2$ are often considered factors influencing insect evolutionary dynamics[64]. For example, the fluctuation in atmospheric $O_2$ has long been hypothesised as the main driver of the decline, in the Permian, of the charismatic giant griffinflies of the family Meganeuridae[65–67]. We found that drops in atmospheric $O_2$ may have slowed down insect diversification during both the Permian and the Triassic. Finally, continental fragmentation is considered a major driver of diversification for some insect lineages, promoting allopatric

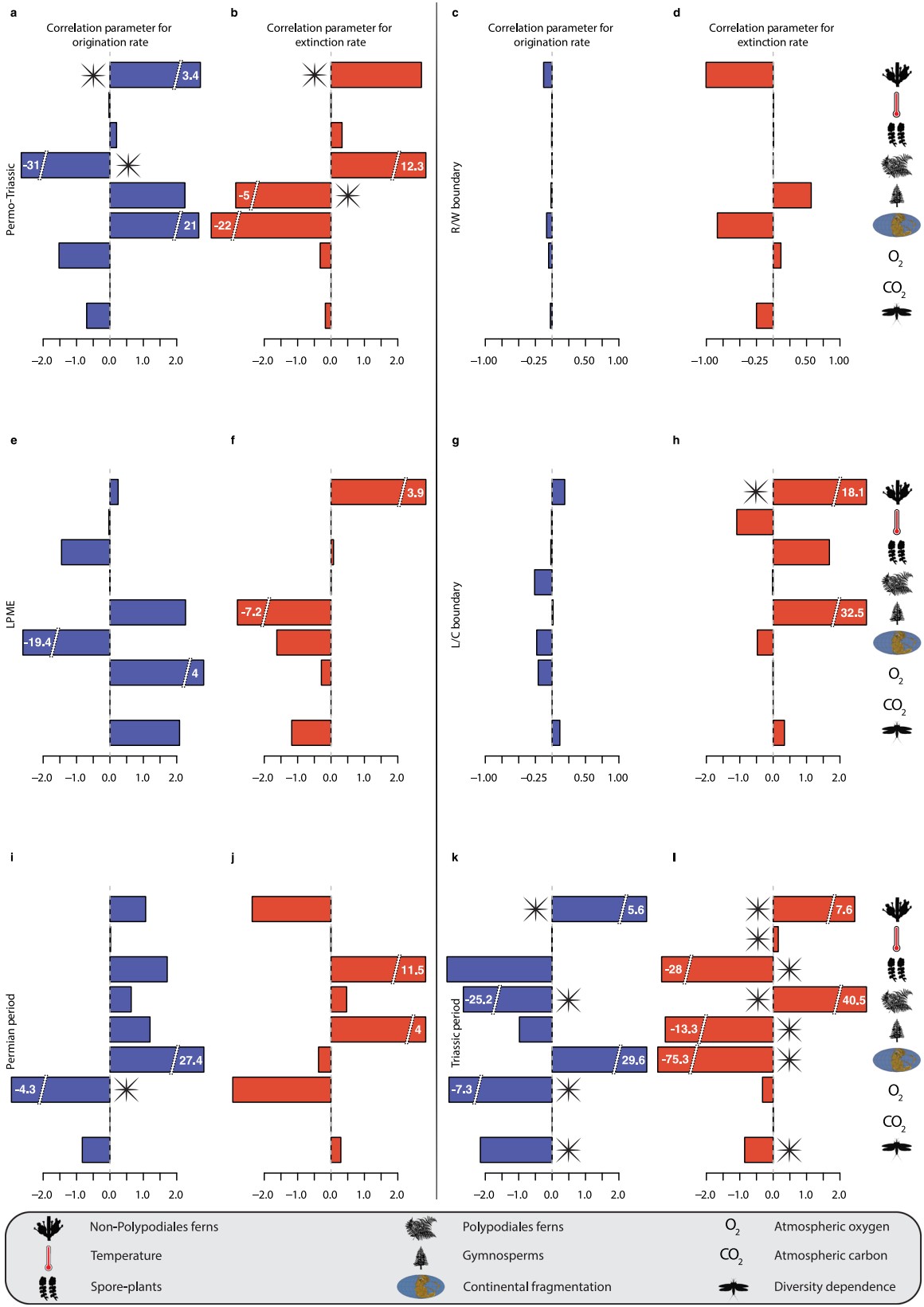

speciation but also deeply altering previous ecosystems[26,68]. This process was very active during the Permo–Triassic with the initiation of the Pangea fragmentation into Laurasia and Gondwana[69]. It has slowed down extinction and accelerated the origination of insects during the Triassic, which suggests an overarching role in allopatric speciation during this period[70].

Importantly, the MBD model not only captured the previously demonstrated diversity declines and increase—negative and positive net diversification rates inferred with BDCS and RJMCMC models—but also revealed that this dynamic could be due to global changes in floral assemblages (Supplementary Fig. 18). Overall, we found a preponderant role of plant diversification and

**Fig. 3 | Paleoenvironmental correlates and potential drivers of the dynamics of insects.** Bayesian inferences of correlation parameters on origination (blue: **a**, **c**, **e**, **g**, **i**, **k**) and extinction (red: **b**, **d**, **f**, **h**, **j**, **l**) with abiotic factors like global temperature, continental fragmentation, global variation of atmospheric $CO_2$ and $O_2$; with biotic factors like relative diversity through time of Polypodiales ferns, non-Polypodiales ferns, Gymnosperms and spore-plants; and diversity-dependence factors with diversity through time of insect genera. Stars indicate a significant correlation parameter for a given variable (shrinkage weights (ω) > 0.5). Colour and abbreviations as in Fig. 1. Silhouettes from http://phylopic.org/. Gnetophyta (by Curtis Clark and T. Michael Keesey), Polypodiales ferns (by Olegivvit) licences at https://creativecommons.org/licenses/by-sa/3.0/; Hepaticae, Gymnosperm, Palaeodictyoptera, licences at https://creativecommons.org/publicdomain/zero/1.0/. Palaeomap (by Eikeskog1225; licence at https://creativecommons.org/licenses/by-sa/4.0/). Other items made by Corentin Jouault.

extinction on the dynamic of insect diversification across the Permo–Triassic.

## Diversity dependence within and between insect guilds

Interactions between and within guilds form a complex network, in which the wax and wane of clades directly influence the intensity and the direction (positive or negative) of interactions, which is known as diversity dependence. Therefore, the diversification or decline of clades within a guild may result in a cascade effect leading to the diversification or extinction of lineages from the same or a distinct guild. Under negative interactions, increasing taxonomic diversity in one clade decreases origination rates and/or increases extinction rates in another clade. For example, herbivorous insects are known to experience high selection pressure from their competitors or predators such that these interactions can affect their dynamic:[71–73] the diversification of predators increases the predation pressure, which could limit speciation and/or increase extinction in prey lineages[31]. Similarly, the modification of predator assemblage is known to affect prey abundance and distribution. Predator interactions have been shown to be triangular, with top predators feeding on meso-predators and on prey[74]. Changes in predator assemblages also affect non-prey taxa and ecosystem processes through indirect pathways[75,76], and may also facilitate the coexistence of prey species (predator-mediated coexistence)[77].

Our MBD analysis ran for all insects without distinction between ecological guilds revealed that within-clade diversity-dependent processes played a role in insect origination and extinction only during the Triassic (Fig. 3k, l and Supplementary Table 8): intra-clade diversity was negatively correlated with origination ($g\lambda = -2.15$, 95% CI = $-2.95$–$-1.43$) and extinction ($g\mu = -0.85$, 95% CI = $-1.73$–$-0.12$). This result means that origination decreased when the global insect diversity increased, suggesting intra-clade diversity dependence. The second result indicates that extinction decreased when insect diversity increased, suggesting positive interactions. This result could mean that diversification increased dependence on resources and particularly for food, limiting origination by intra-clade negative interactions[61]. Oppositely, in a changing environment such as during the Permo–Triassic, the diversification of insects allowed the colonisation of novel ecological niches that resulted from the wax and wane of plants but also from the fragmentation of continents[70,78].

These results are cross-validated by our MBD analyses that also investigate, in parallel to abiotic factors, diversity dependence between and within the three guilds (Supplementary Tables 9–11). We notably recorded intra-clade diversity dependence in herbivores ($g\lambda = -2.16$, 95% CI = $-3.71$–$-0.81$; Supplementary Table 10) and the negative interactions between herbivores and 'others' insects ($g\mu = 0.25$, 95% CI = $0.17$–$0.32$; Supplementary Table 10).

The MCDD analyses allowed an in-depth study of the diversity-dependent interactions within and between clades. It revealed recurrent negative and positive interactions between guilds and disentangled the interactions influencing insect diversification. In the two MCDD analyses (i.e. with three or four guilds), we found that herbivores hold a central position in the interaction network of the Permo–Triassic (Fig. 4b, Supplementary Fig. 24 and Supplementary Tables 12, 13). Our analyses showed intra-clade diversity dependence, with the increase of diversity within the herbivores resulting in higher extinction rates for this guild ($g\mu = 0.0789$ and $0.0514$ for the analyses with three and four guilds, respectively; Fig. 4b and Supplementary Tables 12, 13). The increase of diversity in generalists and detritivore/fungivore insects (or the combination of these two guilds) also results in higher extinction rates in herbivores (Fig. 4b and Supplementary Tables 12, 13). These interactions are also observed in extant fauna with extant phytophagous insects competing for plants and sometimes avoiding plants already colonised by other phytophagous insects[79,80]. For example, because of the similarity of their mouthparts and feeding strategies, Hemiptera might have outcompeted Palaeodictyopteroidea, likely contributing to their extinction[56,81,82]. The increase of diversity in predators results in lower extinction rates in herbivores ($g\mu = -0.2054$ and $-0.1434$ for the analyses with three and four guilds, respectively; Fig. 4b and Supplementary Tables 12, 13). This result could be interpreted as predator-mediated coexistence, a well-known and commonly observed interaction between extant clades[83,84]. Recording a similar effect through deep time is unsurprising if one assumes that trophic interactions were maintained through time, the sole difference being the absence of parasitoids.

The central position of herbivores is also found for origination rates, with the increase of diversity in the generalists and detritivores/fungivores (or the combination of these two guilds) correlated with higher origination in herbivores (Supplementary Tables 12, 13). Our analyses indicated that herbivores benefited from the increased diversity of 'others' insects (Fig. 4b). We interpret these results such that increasing the diversity of one guild led to a decrease in predation pressure on the others, likely by diversifying the possible prey choices for predators. Alternatively, it could also result from 'diversity begets diversity' and 'community-level defence' mechanisms[85–87].

We also recorded biotic interactions between predators and 'others' insects (i.e. non-herbivorous and non-predators). The increase in predator diversity correlated with lower origination of the 'others' insects ($g\lambda = 0.0118$; Fig. 4b and Supplementary Table 13). Finally, the increase in the diversity of the "others" insects correlated with higher extinction rates for this group ($g\mu = 0.0302$; Fig. 4b and Supplementary Table 13). Therefore, it is likely that these "others" insects were impacted by predation during the Permo–Triassic, but also that intra-clade diversity dependence occurred within this group.

Complex interactions were arguably already established between and within insect ecological guilds during the Permian and Triassic (Fig. 4b). Specifically, herbivores were strongly interacting with other insect guilds, either by negative interactions or facilitation. This mirrors the trophic interactions recorded in modern ecosystems, wherein herbivores, by their abundance, their role as prime consumers and their interactions with their predators and competitors, contribute to shape the diversity of other insects.

## Limitations

We have investigated the past dynamic of insect diversification based on the fossil record and Bayesian inferences. Although powerful, this approach has limitations either related to the dataset or to the analytical methods. First, a major issue when studying fossil insects is the poor delineation of several clades, at all systematic levels (e.g. order: 'Grylloblattodea', genus: *Geinitzia* Handlirsch). Some lineages serve as taxonomic wastebaskets and their composition greatly varies depending on the authors[88,89]. This limitation is particularly important

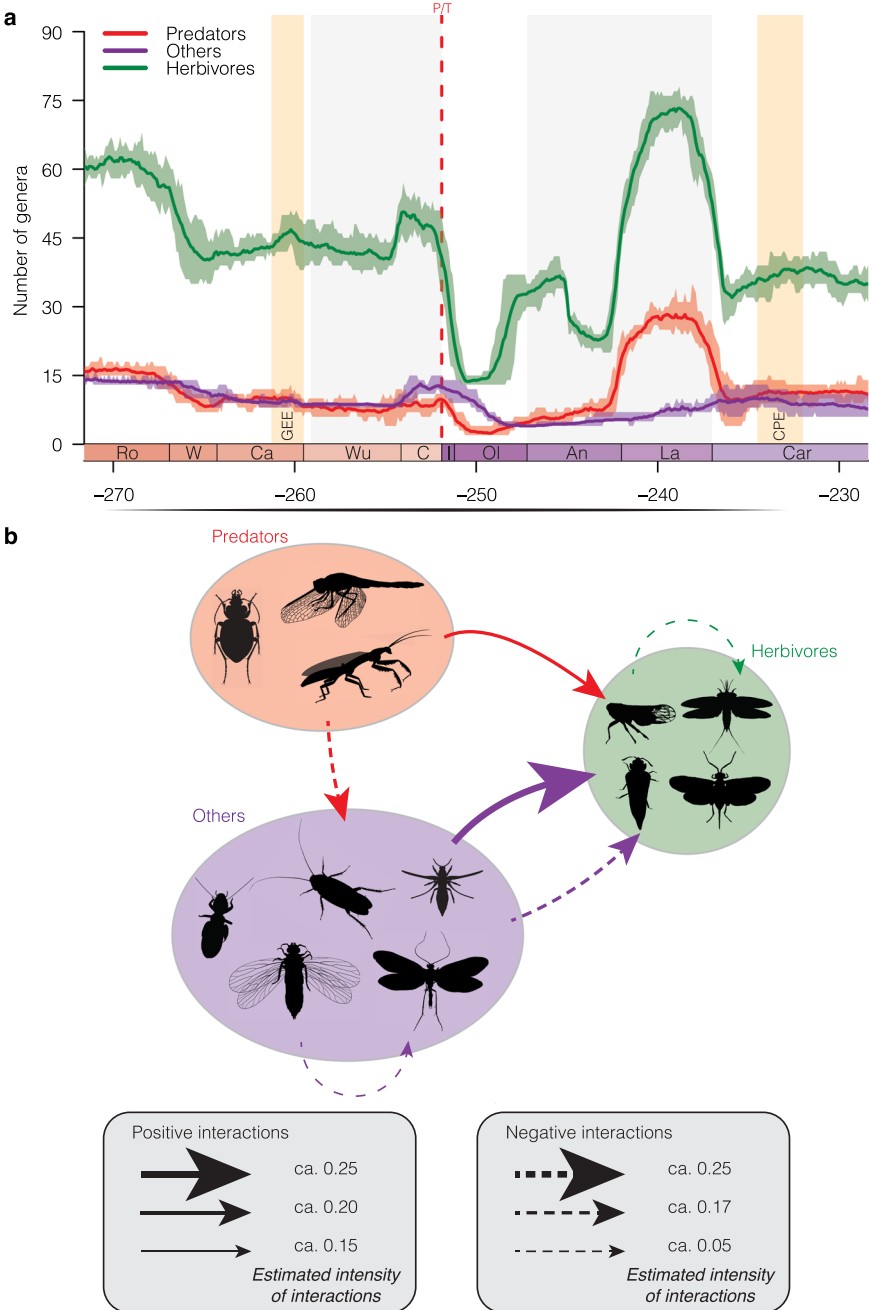

**Fig. 4 | Diversity trajectories and the effect of diversity dependence or facilitation for guilds of insects. a** Diversity trajectories of three guilds of insects between Roadian and Carnian. Reconstructions of diversity trajectories replicated ten times, incorporating uncertainties around the ages of the fossil occurrences. For each plot, solid lines indicate mean posterior rates; shaded areas show 95% CI. **b** Network showing positive and negative interactions within and between guilds (only significant correlations are shown). Each arrow indicates the intensity of interaction imposed by a given guild toward another one. Colour and abbreviations as in Fig. 1. Silhouettes from http://phylopic.org/. Titanoptera (by Melissa Broussard), and Trichoptera (by Didier Descouens; vectorized by T. Michael Keesey) licences at https://creativecommons.org/licenses/by-sa/3.0/; Coleoptera, Dictyoptera, Hemiptera, Hymenoptera, Odonatoptera, Mecoptera, Miomoptera, Palaeodictyoptera, Plecoptera and Psocodea licences at https://creativecommons.org/publicdomain/zero/1.0/; Dermaptera, Thysanoptera and Coleoptera by Corentin Jouault.

for fossil insects from the Carboniferous to Jurassic deposits, maybe because they likely belong to extinct major clades or to stem groups of modern lineages and share characters with different orders[23]. To limit this issue, we did not analyse these groups at the level known to be problematic (i.e. order) and instead included the occurrences at a higher systematic level (see Methods). Future discoveries will hopefully clarify these problematic groups and help refine the inferred diversification patterns. Second, some lineages are underrepresented in the fossil record at any period because of their biology or their morphology (e.g. Zoraptera and Lepidoptera[90]). This taphonomic bias is well known for some groups and may hamper our understanding of the past dynamic of insects. We cannot overcome this limitation, but new discoveries should also bring additional information on the past diversity of fossil and modern lineages. Note that PyRate correctly evaluates past dynamics with incomplete taxon samplings[31,38]. Third, as with any process-based model, PyRate makes assumptions about the processes generating the evolutionary history of a clade. These assumptions can violate real evolutionary processes. For instance, we

did not account for any geographic bias in sampling fossils while modelling the diversification rate. Yet, fossil insect deposits are mainly known from the northern latitudes, while the southern hemisphere only accounts for five major insect Lagerstätten and only two during the Permo–Triassic[23]. Fourth, our results depend on the choice and availability of environmental and biological variables used as putative diversification drivers. We focused on eight candidates (non-related to diversity dependence) reflecting widespread environmental changes as likely factors that could have influenced the diversification of insects. Additional factors could be at play, such as volcanism or changes in rainfall frequency and intensity[12] and would deserve attention in the future. Fifth, the ecology of numerous fossil insects is unknown or hypothetical, hampering our understanding of biotic interactions in the deep past. Similarly, because the ecology is simplified into broad categories[91,92] or sometimes unknown, conclusions can be flawed. Nonetheless, because of the size of our dataset and the conservatism of our investigation, our results should be relatively robust and provide testable conclusions. Finally, the number of fossil specimens is often considered a limitation for the study of insect past dynamics[23,93]. We have integrated all fossil insect occurrences, following an in-depth curation of the fossil insect literature, for a longer time interval than the period of interest, which limits the edge effect and creates a sufficient background for the model to estimate diversification shifts in the period of interest. This approach accounts for all currently known information on fossil insects and is more likely to reflect past insect diversity dynamics.

## Concluding remarks

Insects have roamed the Earth since the Devonian, but our results show that they experienced recurrent periods of extinction and diversification during the Permo–Triassic. They underwent substantial extinctions around the Roadian–Wordian, the LPME and the Ladinian–Carnian boundaries. Those crises had a heterogeneous effect on insect groups, which advocates for a cautious interpretation of global extinction patterns. A biological explanation for variations in diversification during the Permo–Triassic is presented, mostly relying on biotic factors, although we acknowledge that our approach accounts for only part of the explanatory variables. Changes in floral assemblages, notably the variations of the diversity of gymnosperms and ferns, have likely played a significant role in the wax and wane of insects. Additionally, diversity-dependence interactions, particularly those involving herbivores, have likely impacted insect extinction or diversification during the Permo–Triassic, with a putative effect of predator-mediated coexistence. This study encourages a complete reinterpretation of the dynamic of insects in deep time. With over 500,000 extant insect species threatened with extinction[94], studying mass extinctions in the deep past and deciphering their triggers are important and timely challenges.

## Methods

### Fossil record of insects

We compiled all species-level fossil occurrences of insects using https://paleobiodb.org/ (PBDB) as a starting point (downloaded October 12, 2021). The dataset obtained from PBDB contained initially 5808 occurrences for a period ranging from the Asselian to the Rhaetian. The dataset was cleaned of synonyms, outdated combinations, *nomina dubia*, and other erroneous and doubtful records, based on revisions provided in the literature and/or on the expertise of the authors. After correction, including data addition from the literature, our dataset was composed of 3636 species (1784 genera, and 418 families) for 17,250 occurrences resulting from an in-depth study and curation of the entire bibliography of fossil insects, spanning from the Asselian (lowermost Permian) to the Rhaetian (uppermost Triassic). Although most of the taxa included in the datasets are nominal taxa (published and named), a few unnamed taxa (genera or species) that

are considered separate from others were also included, although not formally named in the literature or not published yet. These unpublished taxa are identifiable by the notation 'fam. nov.' or 'gen. nov.' following their names.

Occurrences used here are specimens originating from a given stratigraphic horizon assigned to a given taxon. The age of each occurrence is based on data from PBDB, corrected with a more precise age (generally stage, sometimes substage), and the age of each time bin boundaries relies on the stratigraphic framework proposed in the International Chronostratigraphic Chart (updated to correspond with the ICS 2022/02[95]). Similarly, the ages of some species assigned to the wrong stage were corrected. In fact, some species from the French Permian deposit of Lodève were initially considered to be of Artinskian age in PBDB but most species from this deposit originate from the Merifons member, which is of Kungurian age[96].

Our data compilation allows a robust integration of data before and after our period of interest (i.e. the lower Permian and all geologic stages after the Carnian) to encompass occurrences of genera that may survive until the Late Triassic and to generate a sufficient background for the model to correctly estimate the extinction events around the P/T boundary. Since we used different datasets, the differences between genus-level or family-level occurrence numbers are explained by the systematic placement of some specimens that can only be placed confidently in a family but not in a genus (Supplementary Table 1). Tentative species identifications originally placed with uncertainty (reported as 'aff.' or '?') were always included at a higher taxonomic level. Uncertain generic attributions were integrated as occurrences at the family level (e.g. a fossil initially considered *Tupus?* is recorded as an occurrence of Meganeuridae). Our total dataset was subdivided into smaller datasets, which represent orders or other subclades of insects (e.g. Mecoptera, Holometabola and Polyneoptera). Note that all the ichnospecies—a species name assigned to trace fossils (e.g. resting trace, nest and leaf damage)— and insect eggs (e.g. *Clavapartus latus*, *Furcapartus exilis* and *Monilipartus tenuis*) were not included in the analyses[97]. To prevent potential issues regarding the diversification estimates for clades with poor delineation, we refrained from analysing several orders that serve as taxonomic 'wastebaskets' (e.g. Grylloblattodea). These groups are poorly defined, likely polyphyletic or paraphyletic, and not supported by apomorphic characters—e.g. the monophyly of the 'Grylloblattodea' (Grylloblattida Walker, 1914 *plus* numerous fossil families and genera of uncertain affinities) is not supported by any synapomorphy, nor the relationships within this group. The occurrences assigned to these orders were rather included in analyses conducted at a higher taxonomic level (at the Polyneoptera level in the case of the 'Grylloblattodea'). The detail of the composition of all the datasets is given in Supplementary Table 14, and each dataset is available in Supplementary Data 1.

Studying extinction should, when possible, rely on species-level diversity to better circumscribe extinction events at this taxonomic rank, which is primarily affected by extinction[98–100]. However, in palaeoentomology, species-level occurrence data may contain less information than genus-level data, mainly because species are most of the time only known from one deposit, resulting in reduced life span, and are also sometimes poorly defined. Insects are also less prone to long-lasting genera or species than other lineages, maybe because of the relatively short time between generations (allowing for rapid evolution) or because morphological characters are better preserved or more diagnostic than in other lineages (i.e. wing venation), allowing easier differentiation. Another argument for the use of genus-level datasets is the possibility to add occurrences represented by fossils that cannot be assigned at the species level because of poor preservation or an insufficient number of specimens/available characters. By extension, the genus life span provides clues as to survivor taxa and times of origination during periods of post-extinction or recovery. A genus encompassing extinction events indicates that at least one

species of this genus crossed the extinction. To get the best signal and infer a robust pattern of insect dynamics around the P/T events, we have chosen to analyse our dataset at different taxonomic ranks (e.g. genus, family and order levels) to extract as much evidence as possible.

To further support our choice to work at these different levels, most recent works aiming to decipher the diversification and extinction in insect lineages have worked using a combination of analyses[21,22,26]; this also applies to non-insect clades[51,101,102]. This multi-level approach should maximise our understanding of the Permo–Triassic events.

## Assessing optimal parameters and preliminary tests

Prior to choosing the settings for the final analyses (see detail in Dynamics of origination and extinction), a series of tests were carried out to better evaluate the convergence of our analyses. First, we analysed our genus-level dataset with PyRate[36] running for 10 million generations and sampling every 10,000 generations, on ten randomly replicated datasets using the reversible-jump Markov Chain Monte Carlo (RJMCMC) model[37] and the parameters of PyRate set by default. As the convergence was too low, new settings were used, notably increasing the number of generations to 50 million generations and monitoring the MCMC mixing and effective sample size (ESS) each 10 million generations. We modified the minimal interval between two shifts (-min_dt option, testing 0.5, 1.5 and 2), and found no major difference in diversification patterns between our tests. We have opted for 50 million generations with a predefined time frame set for bins corresponding to the Permian and Triassic stages, and a minimum interval between two shifts of two Ma. These parameters allow for maintaining a short bin frame and high convergence values while correctly identifying periods of diversification and extinction. For each analysis, ten datasets were generated using the *extract.ages* function to randomly resample the age of fossil occurrences within their respective temporal ranges (i.e. resampled ages are randomly drawn between the minimum and the maximum ages of the geological stratum). We monitored chain mixing and ESS by examining the log files in Tracer 1.7.1[103] after excluding the first 10% of the samples as a burn-in period. The parameters are considered convergent when their ESS are greater than 200.

## Dynamics of origination and extinction

We carried out the analyses of the fossil datasets based on the Bayesian framework implemented in the programme PyRate[36]. We analysed the fossil datasets under two models: the birth–death model with constrained shifts (BDCS[38]) and the RJMCMC (-A 4 option[37]). These models allow for a simultaneous estimate for each taxon: (1) the parameters of the preservation process (Supplementary Fig. 17), (2) the times of origination (*Ts*) and extinction (*Te*) of each taxon, (3) the origination and extinction rates and their variation through time for each stage and (4) the number and magnitude of shifts in origination and extinction rates.

All analyses were set with the best-fit preservation process after comparing (-PPmodeltest option) the homogeneous Poisson process (-mHPP option), the non-homogeneous Poisson process (default option), and the time-variable Poisson process (-qShift option). The preservation process infers the individual origination and extinction times of each taxon based on all fossil occurrences and on an estimated preservation rate, denoted *q*, expressed as expected occurrences per taxon per Ma. The time-variable Poisson process assumes that preservation rates are constant within a predefined time frame but may vary over time (here, set for bins corresponding to stages). This model is thus appropriate when rates over time are heterogeneous.

We ran PyRate for 50 million MCMC generations and a sampling every 50,000 generations for the BDCS and RJMCMC models with time bins corresponding to Permian and Triassic stages (-fixShift option). All analyses were set with a time-variable Poisson process (-qShift option)

of preservation and accounted for varying preservation rates across taxa using the Gamma model (-mG option), that is, with gamma-distributed rate heterogeneity with four rate categories[36]. As explained above, the minimal interval between two shifts (-min_dt option) was modified and a value of 2 was used. The default prior to the vector of preservation rates is a single gamma distribution with shape = 1.5 and rate = 1.5. We reduced the subjectivity of this parameter, and favoured a better adequation to the data, allowing PyRate to estimate the rate parameter of the prior from the data by setting the rate parameter to 0 (-pP option). Therefore, PyRate assigns a vague exponential hyper-prior to the rate and samples the rate along with all other model parameters. Similarly, because our dataset does not encompass the entire fossil record of insects, we assumed that a possible edge effect may interfere with our analyses, with a strong diversification during the lowermost Permian and, conversely a strong extinction during the uppermost Triassic. Because the RJMCMC and BDCS algorithms look for rate shifts, we constrained the algorithm to only search for shifts (-edgeShift option) within the following time range 295.0 to 204.5 Ma. We monitored chain mixing and ESS by examining the log files in Tracer 1.7.1[103] after excluding the first 10% of the samples as a burn-in period. The parameters are considered convergent when their ESS are greater than 200.

We then combined the posterior estimates of the origination and extinction rates across all replicates to generate rates through-time plots (origination, extinction, and net diversification). Shifts of diversification were considered significant when log Bayes factors were >6 in the RJMCMC model, while we considered shifts to be significant in the BDCS model when mean rates in a time bin did not overlap with the 95% credibility interval (CI) of the rates of adjacent time bins.

We replicated all the analyses on ten randomly generated datasets of each clade and calculated estimates of the *Ts* and the *Te* as the average of the posterior samples from each replicate. Thus, we obtained ten posterior estimates of the *Ts* and *Te* for all taxa and we used these values to estimate the past diversity dynamics by calculating the number of living taxa at each time point. For all the subsequent analyses, we used the estimated *Ts* and *Te* of all taxa to test whether or not the origination and the extinction rate dynamics were correlated with particular abiotic factors, as suggested by the drastic changes in environmental conditions known during the Permo–Triassic. We used proxies for abiotic factors, such as global continental fragmentation or the dynamic of major clades of plants, and for biotic factors via species interaction within and between ecological guilds. This approach avoids re-modelling preservation and re-estimating times of origination and extinction, which reduces drastically the computational burden, while still allowing to account for the preservation process and the uncertainties associated with fossil ages. Similarly, the times of origination and extinction used in all the subsequent analyses were obtained while accounting for the heterogeneity of preservation, origination and extinction rates. To discuss the magnitude of the periods of extinction and diversification, we compared the magnitude of these events to the background origination and extinction rates (i.e. not during extinction or diversification peaks).

The PyRate approach has proven to be robust following a series of tests and simulations that reflect commonly observed biases when modelling past diversity dynamics[31,38]. These simulations were based on datasets simulated under a range of potential biases (i.e. violations of the sampling assumptions, variable preservation rates, and incomplete taxon sampling) and reflecting the limitations of the fossil record. Simulation results showed that PyRate is able to correctly estimate the dynamics of origination and extinction rates, including sudden rate changes and mass extinction, even if the preservation levels are low (down to 1–3 fossil occurrences per species on average), the taxon sampling is partial (up to 80% missing) or if the datasets have a high proportion of singletons (exceeding 30% of the taxa in some cases).

The strongest bias in birth–death rate estimates is caused by incomplete data (i.e. missing lineages) altering the distribution of taxa; a pervasive effect often mentioned for phylogeny-based models[104–106]. However, in the case of PyRate, the simulations confirm the absence of consistent biases due to an incomplete fossil record[36]. Finally, the recently implemented RJMCMC model was shown to be very accurate for estimating origination and extinction rates (i.e. more accurate than the BDCS model, the boundary-crossing and three-time methods) and is able to recover sudden extinction events regardless of the biases in the fossil dataset[37].

## The severity of extinctions and survivors

For each event—the Roadian–Wordian, the LPME, and the Ladinian–Carnian—we quantified the percentage of extinctions and survivors at the genus level. We used the $Te$ and $Ts$ from our RJMCMC analysis and computed the mean for the $Te$ ($Te_m$) and for the $Ts$ ($Ts_m$) of each genus. We then filtered our dataset to keep only the genera with a $Ts_m$ older than the upper boundary of the focal event, i.e., we only kept the genera that appeared before the end of the event. Then, we discarded the genera that have disappeared before the lower boundary of the focal event, i.e. $Te_m$ older that the lower boundary of the event. The remaining genera, which corresponds to all the genera (total) present during the crisis ($Tt_{gen}$), can be classified into two categories, 'survivor genera' ($S_{gen}$), i.e. those that survived the crisis, and those that died: 'extinct genera' ($E_{gen}$). The survivors have a $Te_m$ younger than the upper boundary of the focal event, while the 'extinct genera' died out during the event and have a $Te_m$ between the lower and upper boundaries of the event of interest. To obtain the percentage of survivors, we used the following formula: $(S_{gen}/Tt_{gen}) \times 100$. Similarly, the percentage of extinction is calculated as: $(E_{gen}/Tt_{gen}) \times 100$.

## Age-dependent extinction model

We assessed the effect of taxon age on the extinction probability by fitting the age-dependent extinction (ADE; -ADE 1 option) model[50]. This model estimates the probability for a lineage to become extinct as a function of its age, also named longevity, which is the elapsed time since its origination. It is recommended to run the ADE model over time windows with roughly constant origination and extinction rates, as convergence is difficult—but not impossible—to reach in extinction or diversification contexts[50]. We ran PyRate for 50 million MCMC generations with a sampling every 50,000 generations, with a time-variable Poisson process of preservation (-qShift option), while accounting for varying preservation rates across taxa using the Gamma model (-mG option). We replicated the analyses on ten randomised datasets and combined the posterior estimates across all replicates. We estimated the shape ($\Phi$) and scale ($\Psi$) parameters of the Weibull distribution, and the taxon longevity in a million years. According to ref. 50, there is no evidence of age-dependent extinction rates if $\Phi = 1$. However, the extinction rate is higher for young species and decreases with species age if $\Phi < 1$, and extinction rates increase with species age if $\Phi > 1$. Although ADE models are prone to high error rates when origination and extinction rates increase or decrease through time, simulations with PyRate have shown that fossil-based inferences are robust[50]. We investigated the effect of ADE during three different periods (-filter option) as follows: (1) between 264.28 Ma and 255 Ma (pre-decline), (2) between 254.5 Ma and 251.5 Ma (decline) and (3) between 234 Ma and 212 Ma (post-crisis). We monitored chain mixing and ESS by examining the log files in Tracer 1.7.1[103] and considered the convergence of parameters sufficient when their ESS were greater than 200.

## Selection of abiotic and biotic variables

To test correlations of insect diversification with environmental changes, we examined the link between a series of environmental variables and origination/extinction rates over a period encompassing the GEE, the LPME and the CPE but also for each extinction event. We focused on the role of nine variables, also called proxies, which have been demonstrated or assumed to be linked to extinctions and changes in insect diversity[26,67].

The variations in the atmospheric $CO_2$ and $O_2$ concentrations are thought to be correlated with the diversification of several insect lineages, including the charismatic giant Meganeuridae[65–67]. Because the increase of $O_2$ concentration has likely driven the diversification of some insects, its diminution may have resulted in the extinction or decline of some lineages. Therefore, we investigated the potential correlation of the variations of this variable with insect dynamics using data from ref. 55. We extracted the data, with 1-million-year time intervals, spanning the Permo–Triassic.

Similarly, the modification of $CO_2$ concentration, notably its increase, is known to promote speciation in some modern insect groups[107]. Therefore, a similar effect may have occurred during the Permian and Triassic but remains to be tested. We based our analyses on the dataset of ref. 108. We used their cleaned dataset and extracted all verified values for the Permo–Triassic interval. Because the initial data (i.e. independent estimates) were made in various locations for the same age, different values of the $CO_2$ concentration are provided. We incorporated all these values in our analysis, allowing PyRate to search for a correlation for each value of the $CO_2$ concentration. We obtain a final correlation independent of the sampling location, in line with our large-scale analysis.

The continental fragmentation, as approximated by plate tectonic change over time, has recently been proposed as a driver of Plecoptera dynamics[26]. Because the period studied encompasses a major geological event, the fragmentation of the supercontinent Pangea, we investigated the effect of continental fragmentation on insect diversification dynamics. We retrieved the index of continental fragmentation developed by ref. 69 using paleogeographic reconstructions for 1-million-year time intervals. This index approaches 1 when all plates are disjoined (complete plate fragmentation) and approaches 0 when continental aggregation is maximal.

Climate change (variations in warming and cooling periods) is a probable driver of diversification changes over the history of insects[21,109]. Temperature is likely directly linked with insect dynamics[109] but also with their food sources, notably plants[110]. Because it was demonstrated that modification of temperature impacted floral assemblages[110], we tested the correlation between temperature variations and the diversification dynamic of insects. Major trends in global climate change through time are typically estimated from relative proportions of different oxygen isotopes ($\delta^{18}O$) in samples of benthic foraminiferan shells[111]. We used the data from ref. 112, converted to absolute temperatures following the methodology described in Condamine et al.[113] (see their section *Global temperature variations through time*). The resulting temperature data reflects planetary-scale climatic trends, with time intervals inferior to 1-million-year, which can be expected to have led to temporally coordinated diversification changes in several clades rather than local or seasonal fluctuations.

The fluctuation in relative diversity of gymnosperms, non-Polypodiales ferns, Polypodiales ferns, spore-plants, and later the rise of angiosperms has likely driven the diversification of numerous insects[57,60,61,114]. Close interactions between insects and plants are well-recorded during the Permian and Triassic[57,60,61]. In fact, herbivorous insects are known to experience high selection pressure from bottom-up forces, resulting from interactions with their hosts or feeding plants[30,72]. Therefore, it appears crucial to investigate the effect of these modifications on the insects' past dynamics. We used the data from ref. 38 for the different plant lineages (all with 1-million-year time intervals). All the datasets for these variables are available in the publications cited aside from each variable or in Supplementary Data 1.

## Multivariate birth–death model

We used the multivariate birth–death (MBD) model to assess to what extent biotic and abiotic factors can explain temporal variation in origination and extinction rates[55]. The model is described in ref. 55, where origination and extinction rates can change through time in relation to environmental variables so that origination and extinction rates depend on the temporal variations of each factor. The strength and sign (positive or negative) of the correlations are jointly estimated for each variable. The sign of the correlation parameters indicates the sign of the resulting correlation. When their value is estimated around zero, no correlation is estimated. An MCMC algorithm combined with a horseshoe prior, controlling for over-parameterisation and for the potential effects of multiple testing, jointly estimates the baseline origination ($\lambda 0$) and extinction ($\mu 0$) rates and all correlation parameters ($G\lambda$ and $G\mu$)[55]. The horseshoe prior is used to discriminate which correlation parameters should be treated as noise (shrunk around 0) and which represent a true signal (i.e. significantly different from 0). In the MBD model, a correlation parameter is estimated to quantify independently the role of each variable on the origination and the extinction.

We ran the MBD model using 20 (for short intervals) or 50 million MCMC generations and sampling every 20,000 or 50,000 to approximate the posterior distribution of all parameters ($\lambda 0$, $\mu 0$, nine $G\lambda$, nine $G\mu$ and the shrinkage weights of each correlation parameter, $\omega G$). The MBD analyses used the $Ts$ and the $Te$ derived from our previous analyses under the RJMCMC model. The results of the MBD analyses were summarised by calculating the posterior mean and 95% CI of all correlation parameters and the mean of the respective shrinkage weights (across ten replicates), as well as the mean and 95% CI of the baseline origination and extinction rates. We carried out six analyses, over: (1) the Permo–Triassic (between 298.9 and 201.3 Ma); (2) the Roadian–Wordian (R/W) boundary (between 270 and 265 Ma), (3) the LPME (between 254.5 and 250 Ma), (4) the Ladinian–Carnian (L/C) boundary (between 240 to 234 Ma), (5) the Permian period (between 298.9 and 251.902 Ma) and (6) the Triassic period (between 251.902 and 201.3 Ma). We monitored chain mixing and ESS by examining the log files in Tracer 1.7.1[103] and considered the convergence of parameters sufficient when their ESS were greater than 200.

## Multiple clade diversity-dependence model

To assess the potential effect of diversity-dependence on the diversity dynamics of three or four insect guilds, we used the multiple clade diversity-dependence (MCDD) model in which origination and extinction rates are correlated with the diversity trajectory of other clades[31]. This model postulates that competitive interactions linked with an increase in diversity results in decreasing origination rates and/or increasing extinction rates. The MCDD model allows for testing diversity-dependence between genera of a given clade or between genera of distinct clades sharing a similar ecology.

We estimated the past diversity dynamics for three (i.e. herbivores, predators, and a guild composed of generalists + detritivores/fungivores dubbed 'others') or four insect groups or guilds (i.e. herbivores, predators, generalists and detritivores/fungivores) by calculating the number of living species at every point in time based on the times of origination ($Ts$) and extinction ($Te$) estimated under the RJMCMC model (see above) (Supplementary Figs. 19–24). We defined our four insect groups with a cautious approach i.e. insect genera, families or orders for which nothing is known about the ecology or about the ecology of their close relatives were not considered for the analysis. For example, no diet was assigned to Diptera, Mecoptera or Glosselytrodea. The ecology of the Triassic Diptera and Permo–Triassic Mecoptera is difficult to establish because extant Diptera and Mecoptera have a wide diversity of ecology. Fossil Mecoptera are also putatively involved in numerous interactions with

plants (species with elongated mouthparts), suggesting a placement in the herbivore group, while other species were likely predators. Therefore, we cannot decide to which group each species belongs. Similarly, nothing is known about the body and mouthparts of the Glosselytrodea, most of the time described based on isolated wings; we did not assign the order to any group. The definition and delineation of insect clades have also challenged the placement of several orders (e.g. 'Grylloblattodea') in one of our four groups. The order 'Grylloblattodea' is poorly delineated and mostly serves as a taxonomic 'wastebasket' to which it is impossible to assign a particular ecology. Finally, genera, species, or families not placed in a higher clade (e.g. *Meshemipteron*, Perielytridae) were not included in the analysis. Oppositely, the guilds 'herbivores' and 'predators' are well defined, and their ecology is evidenced by the morphology of their representatives and the principle of actualism. For example, the ecology of *Meganeurites gracilipes* (Meganeuridae) has been deeply studied, and its enlarged compound eyes, its sturdy mandibles with acute teeth, its tarsi and tibiae bearing strong spines, and the presence of a pronounced thoracic skewness are specialisations today found in dragonflies that capture their prey while in flight[115]. All Odonatoptera are well-known predator insects. The raptorial forelegs of the representatives of the order Titanoptera and their mouthparts with strong mandibles are linked with predatory habits[81]. The Palaeodictyopteroidea were herbivorous insects with long, beak-like, piercing mouthparts, and probably a sucking organ[81,82]. Most Hemiptera are confidently considered herbivorous insects by comparison with their extant representatives. For example, the Cicadomorpha or Sternorrhyncha are known to feed on plants and their fossil representatives likely possessed the same ecology because of similar morphologies[116]. Some hemipteran families (e.g. Nabidae) are predators and we cautiously distinguished herbivorous and carnivorous taxa among Hemiptera. The detail of the ecological assignations for the 1009 genera included in our analyses can be found in Supplementary Data 1 (Table MCCD).

We calculated ten diversity trajectories from the ten replicated analyses under the RJMCMC model. The estimation of past species diversity might be biased by low preservation rates or taxonomic uncertainties. However, such trajectory curves are likely to provide a reasonably accurate representation of the past diversity changes in the studied clades, notably because the preservation during the Permian and Triassic period is relatively good for insects (i.e. no gaps).

Our MCDD analyses comprise all the insect genera spanning from the lowermost Permian to the uppermost Triassic and were run and repeated on ten replicates (using the $Te$ and $Ts$ estimated under the RJMCMC model) with 50 million MCMC generations and a sampling frequency of 50,000. For each of the four insect groups, we computed the median and the 95% CI of the baseline origination and extinction rates ($\lambda i$ and $\mu i$), the within-group diversity-dependence parameters $g\lambda i$ and $g\mu i$, and the between-groups diversity dependence parameters $g\lambda ij$ and $g\mu ij$. The mean of the sampled diversity dependence parameters (e.g. $g\lambda ij$) was used as a measure of the intensity of the negative (if positive) or positive interactions (if negative) between each pair of groups. The interactions were considered significant when their median was different from 0 and the 95% CI did not overlap with 0. We monitored chain mixing and ESS by examining the log files in Tracer 1.7.1[103] and considered the convergence of parameters sufficient when their ESS were greater than 200.

We cross-validated the result of the MCDD model using the MBD model. The MBD model can be used to run a multiple clade diversity-dependence analysis by providing the diversity trajectories of insect guilds as a continuous variable. These data are directly generated by PyRate using the lineages-through-time generated by the RJMCMC analyses (-ltt option). We ran the MBD model using 50 million MCMC generations and sampling every 50,000 to approximate the posterior distribution of all parameters ($\lambda 0$, $\mu 0$, four $G\lambda$, four $G\mu$ and the

shrinkage weights of each correlation parameter, ωG). We carried out three analyses, over the period encompassing the three extinction events (between 275 and 230 Ma): (1) for herbivores; (2) for predators; and (3) for 'others'. For each analysis, the lineages-through-time data of the two other guilds are used as continuous variables to investigate a diversity dependence effect. We monitored chain mixing and ESS by examining the log files in Tracer 1.7.1[103] and considered the convergence of parameters sufficient when their ESS were greater than 200.

### Reporting summary
Further information on research design is available in the Nature Portfolio Reporting Summary linked to this article.

## Data availability
The data generated in this study are provided in Supplementary Data 1. The datasets used for the RJMCMC and BDCS analyses, all the guilds assignments for the MCDD analyses, all the paleoenvironmental variables used for the MBD model, and all the references used to build the different datasets can be found in Supplementary Data 1 and have been deposited in the Figshare digital data repository (https://doi.org/10.6084/m9.figshare.c.6296196.v2). Source data are provided as a Source Data file. Source data are provided with this paper.

## Code availability
The command lines set to run all the models presented in this study are available in the Supplementary Code or through the Figshare digital data repository (https://doi.org/10.6084/m9.figshare.c.6296196.v2).

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

## Acknowledgements

We are grateful to the PBDB team and contributors for their constant effort in maintaining their database and for making it freely available. We thank Dr. Daniele Silvestro for assistance with PyRate. This work is part of the PhD project of C.J.

## Author contributions

C.J., F.L. and F.L.C. designed and conceived the research. C.J. assembled the fossil data with discussions with A.N. C.J. and F.L.C. analysed the data. All authors contributed to the interpretation and discussion of results. C.J. drafted the paper with substantial input from all authors.

## Competing interests

The authors declare no competing interests.
