## [Peer Review File · Nature Communications]

Multiple drivers and lineage-specific insect extinctions during the Permo–TriassicREVIEWER COMMENTS

Reviewer #1 (Remarks to the Author):

The paper by Jouault and colleagues entitled “Insect extinctions during the Permo-Triassic were driven by changes in floral assemblages and guilds’ interactions” represents a valuable contribution to the knowledge on the impact of the End-Permian Mass Extinction event on insects, and more generally on terrestrial invertebrates. In this paper, the authors performed extensive analyses relying on Bayesian inference exploiting the available insects fossil record and provide support for at least three events of extinctions between the Guadalupian and the Middle/Upper Triassic. The results in terms of origination/extinction rates of insect lineages inferred for the genus level almost reflect those obtained by Nicholson et al. 2015 for the family level using different datasets (doi:10.1371/journal.pone.0128554). The manuscript is well written, and the adopted methodologies are robust and appropriate. One of my minor notes concerns the fact that in the Results & Discussion there are several sentences providing information on the Methods, as well as some sentences that could be moved in the introduction. Thus, I suggest considering moving these sentences to the Methods section (e.g., lines 199-209) and Introduction section (e.g., lines 193-199). See also lines from 280 to 297 in the paragraph “Competition and facilitation within and between insect guilds”. Supplementary information, in some cases, should be improved in order to provide more detailed information (e.g. units of CO₂, O₂ concentrations). It seems that authors are not making publicly available the code for the performed analyses, it could be useful to allow repeating the adopted pipeline, but it will also help to understand the strength of the results.

Regarding the first two Results & Discussion sections, I have no concerns, while regarding the section “Ecological drivers of insect diversification and decline”, which in my view represents one of the two novelties presented in this manuscript, I have some doubts regarding the adopted methodology. First, the sentences in the Methods section “To test causal mechanisms of insect diversification ...” (line 596) is in my opinion misleading since to my knowledge the authors have inferred correlations, they are not legitimate to deduce cause-effect relationships. This aspect has to be considered even during the discussion of the achieved results. Furthermore, it is not clear if the authors have analysed the communities considering the different paleoenvironments in which they were established and if they have considered that these communities could differentially react to the same bioclimatic parameters. In addition, the estimates of the bioclimatic parameters were obtained, I guess, from deposits which differ from those where the fossils were dug out. Is it thus possible to expand on a global scale these local paleoclimatic data? Looking at the supplementary excel file - sheets atmospheric concentration of CO₂ and O₂, and Temperature (tables titles are not provided) – it is possible to note that in some cases, as for O₂, data are available with an interval of one million year. Could the author specify how they were estimated since it seems that values in the original papers have no regular intervals? Have the authors interpolated the values of two neighbouring measures? The same consideration could also be done for the temperature. In my view, the authors have to clarify the adopted strategy in the Methods section. In the case of CO₂ are present different values for the same million year (please specify the unit), which in some cases are quite different; how the author managed these measures? E.g. Age = 242.1 and CO₂ = 284, 244, 605. Are these values obtained from deposits characterized by different paleoenvironments? Have the authors considered these values as replicates? All the previously highlighted points make me doubtful about the strength of the achieved results and the inferred considerations.

The paragraph “Limitations” in my view represents a highly appreciable addition, especially the one regarding the trophic guilds to which each taxon belongs.

Minor comments:

Lines 413-414. The fossil dataset used in the analyses includes occurrences spanning from Asselian to the Rhaetian while the bioclimatic parameters, used in the analyses, have a different time-span. Could you please explain this point?

The dataset retrieved by PBDB increased threefold after the addition of occurrences, non-included in PBDB, but published in the scientific literature. I suggest the addition of a supplementary table where the used literature is reported.

Line 422. Please provide information on the unpublished taxa.

Lines 437-438. Please specify on which basis the author decided to attribute tentative species identifications characterized by “aff” or “?” to a higher taxonomic level. In the present version, authors reported “Tentative species identifications ... were most of the time included at a higher taxonomic level”.

Lines 321-324. As an alternative, this pattern could be explained as an indirect effect of an increased amount of plant biomass or other not considered factors.

Reviewer #2 (Remarks to the Author):

The manuscript by Condamine et al. uses existing family- and genus-level records of fossil insects across the Asselian (lowermost Permian) to the Rhaetian (uppermost Triassic) periods from an existing database (<https://paleobiodb.org/>; PBDB) representing 3,636 species (1,784 genera, 418 and 418 families) and 17,250 total detections, after filtering. They use a suite of Bayesian estimation procedures using the process-modeling software, PyRate to estimate changes in diversity and diversification and extinction rates for genera and families for all insects and for each major ancestral clade. Finally, they cross-correlate changes in diversity (and rates therein) with various hypothesized drivers of change, including biotic, abiotic, vicariance events, and "ecological interactions."

The submitted manuscript is generally well-written and reports some interesting trends. I cannot evaluate all the specifics of modeling parameter choice, etc. as I am not specifically familiar with the PyRate program. However, the Methods were convincing and the stated choices appeared defensible.

The Results were interesting and relevant overall. However, I found myself quite unconvinced with the framework and interpretation of "ecological interactions" as drivers of extinction/diversification, which was a major thrust of the paper (even appearing in the title). Ultimately, it is just too big of a stretch to interpret broad correlations among lineages belonging to coarse-scale feeding guilds as the author have done (neg. correlations as evidence of competition and positive as facilitation). For example, surely herbivores might respond negatively to major eruptions while detritivores (for a while, anyway) might benefit). However, this would be classed as competition in the current MS, which is quite unlikely (even impossible since the niches are distinct by definition) in this case. Unless the authors can make a much, much stronger case for the validity of these assumptions (and I don't think they can) I would suggest that work is unpublishable unless removed. This does remove a not-insignificant component of the manuscript but my feeling is that the remaining results have sufficient value to stand alone, though I am not 100% certain of their novelty.

While the MS is well-written and structured, it is hard to get through and confusing in places. Some things that might help include:

- A figure or table with all the date estimates (which is available not until L656 in the Methods). This would make keeping track of the various epochs and transitions much easier for the general reader.
- An Info box or glossary to outline the different insect taxonomic groupings you are using (and perhaps a short explanation as to why for each).
- Explanatory text to accompany the icons, especially in Fig. 3.

I'm confused by Fig. 1 -- what is driving rapid and non-trivial changes in lineage richness (1F) when no such changes are apparent in D and E? D and E are model outputs I gather, but this discrepancy makes me wonder if the lack of corresponding ups and downs reflects a sample size artifact rather than the lack of a pattern that roughly conforms what is seen for genera. Suggest clarifying for readers.

Also, what is the correct scale from the perspective of both time and magnitude for consideration of

impacts on diversity? For example, does the definition of a major extinction include how much net diversification rate dips below zero and for how long. How far below zero does it need to go, and for how long? Does a subsequent recovery influence the interpretation of such an event, or for that matter, might a downturn in diversification following a period of elevated diversification (as appears to be the case for the P/T and L/C event) be better considering a correction? Given the magnitudes of these two events, I'm not really suggesting this latter point as a viable explanation, but what is the interpretation of the spikes before the drops?

Specific comments:

Throughout MS: Diversity is a specific term in ecology that generally combines the concepts of species richness and their relative abundances (evenness). I think most of the many places the word "diversity" is used, "richness" might be better. From my perspective, it would be best to make this change throughout. However, I recognize that terminology usage can vary across sub-disciplines, but please consider.

L79: "genera and at the stage- or formation-level for taxon ages." Something is wrong with this sentence.

L84: "co-occurring guilds" Remove "co-occurring"

L98: "diversity dynamic" Prefer "insect generic richness", "diversification rate", or other term -- "diversity dynamic" doesn't work in this context. Diversity and/or rates can go up or down, but a dynamic just that -- it can't decline as you've stated here. I see that this usage is consistent throughout the MS - I would strongly recommend changing (but will not comment at every usage).

L84: "We simultaneously assessed the effect of co-occurring guilds (herbivores, predators, detritivores/ fungivores, and generalists) on their speciation and extinction rates by quantitatively investigating the roles of competition among insect clades throughout the Permian and Triassic periods. " While I think it's very interesting to consider how major extinction events have manifested similar v. uniquely patterns across these broad feeding guild categories, I am again very dubious that using such a coarse filter can tell you much about positive or negative interactions, or their importance in speciation and extinction dynamics.

L106 (and throughout): "ca." I think you should state these rates without saying "ca." each time. It is implicit that these are estimates. Providing some estimates of uncertainty (i.e., 95% CI range) would be useful though.

L106: "events/ Ma/ lineage" I assume that you mean "events/million years/lineage" in which case "Ma" is not correct.

L181: " However, during the LPME, age had a strong effect on extinction (? = 9.1677, 95% [CI] = 2.3886-18.7343; Supplementary Table S2). " I actually find this compelling but you said in the paragraph above that model convergence is difficult to reach during unstable periods. Could this finding be an artifact of violating the assumptions of stability?

L196: "Insects' past dynamic is inevitably linked to environmental changes, which directly led to their diversification or extinction" This statement is too strong. Sure, environments matter, but vicariance events and novel associations with host plants could be at least as strong a driver in the case of diversification, as is borne out in Fig. 3.

L208: "and four abiotic variables (global temperature variations; global variation of atmospheric CO₂ and O₂; and continental fragmentation)" I assume that you mean temporal (and not spatial) variation (not plural), but if so, at what temporal scale did you calculate? This matters. It also might be appropriate to test for lag effects.

L227: "fluctuation in the diversity of non-Polypodiales would have accelerated their extinction (Fig. 3A, B)." Not "fluctuation" but "increase", no? But I'm confused, aren't these diversity categories overlapping (i.e., Gymnosperm diversity is a subset of non-Polypodiales diversity). It seems more likely that this refers to non-Polypodiales ferns, but this is not clear.

L260: "The amber production of the Carnian might also reflect the beginning of resin production as a defense strategy against phytophagous insects, maybe following the rise of modern insects⁶⁶. " Does this bias the database toward detection of gymnosperm-associated taxa?

L288: "Similarly, detritivores/ fungivores and generalists may compete with herbivores" What is a generalist insect in this context? Usually, they are herbivorous but with fewer restrictions on host plant breadth. This needs to be clarified. Either way, I think it's a major stretch to hypothesize competition among highly distinct feeding guilds. By what mechanism?

L303: ", suggesting intra-clade competition" Big leap to get to intra-clade competition, which I don't really buy. The pattern is interesting though. From L303-348 is where the MS really lost me in terms of its (indefensible, in my opinion) interpretation of cross-correlations as evidence of ecological interaction. I would strongly advocate for deletion or major reworking of these paragraphs.

L441: "ichnospecies" Define

L444: "taxonomic 'bins'." I would explain what you mean here.

L494: "(1) the parameters of the preservation process" Explain.

L506: "Therefore" Is the "Therefore," needed?

L624: "absolute temperatures " Wouldn't variation or change in T be more useful?

Reviewer #3 (Remarks to the Author):

Dear Authors:

This paper represents a great and valuable contribution to diversity dynamics of insects during Permian–Triassic times. The manuscript investigates and describes several important biotic and abiotic factors for explain the diversity of insects during Permian and Triassic. Also, this contribution is very important because never was investigated the diversity decline of insects to genus level, as you commented in the manuscript.

This paper merits publication because builds on previous research and well-developed methodologies, the objectives are clear, the methods used were appropriate, sound, and employed correctly. The multiples statistical test used are concordant with the results, the authors interpret general pattern with caution.

The authors made an exhaustive revision of available information on fossil insects.

The paper is generally well-organized and well-written (the authors use grammar and syntax correctly), but please, bear in mind I am not a native English speaker.

The references are adequate, current and pertinent. All illustrations are necessary and are referred to in the body of the text.

To summarize this is a very interesting paper in all regards. Some minor corrections/suggestions:

- Line 42: replace colloquially called by commonly named.
- Line 46: after warming add a comma (,)
- Line 48: after e.g. add a comma (e.g.,). Please check in all text.
- Line 102: replace (Fig. 1–F) by (Fig. 1D–F).
- Line 136: replace old orders by ancient orders.
- Line 136: after i.e. add a comma (i.e.,). Please check in all text.
- Line 160, Line 695: replace well known by well-known.
- Line 160: replace and the pace of species description by but the description of the species is comparatively slow...
- Line 161 replace Cretaceous or the Cenozoic descriptions by Cretaceous and Cenozoic ones.
- Line 201: replace Permian-Triassic by Permian–Triassic. Please check in text the em-dash.
- Line 217: replace diversities by diversity.
- Line 217-218: replace Polyptodiales have significantly affected insect diversification by ...and non-Polyptodiales significantly affected to insect diversification...
- Line 253: replace concurs with by agree with or coincide with.
- Line 458: replace or of an insufficient by or insufficient number of...
- Line 464: replace signal by evidence
- Line 562: delete the in (the Roadian-Wordian, the LPME, and the Ladinian-Carnian) and replace - by –
- Line 627: replace well recorded by well-recorded
- Line 677-638: replace time through correlations with environmental variables by in relation to environmental variables
- Line 688: replace buccal pieces by mouthparts
- Line 718: add to after thank
- Line 722: add comma after A.N and C.J.

I hope the authors work on the few suggestions and get this paper published soon because it is excellent contribution to Palaeoentomology. Let me know if there is anything else I can help with. Best regards.

Reviewer #4 (Remarks to the Author):

Comments to the authors

This study estimates if the magnitude that well-known extinction events had in insect's diversification. Also, they test if a number of biotic and abiotic factors promoted speciation or extinction rates within the different subclades of Insecta. Finally, they evaluate if feeding strategies are correlated with speciation/extinction rates and how the diversity of each strategy affects the rates of the other strategies.

As a result, they found that the well-known mass extinction events affected insect's diversification heterogeneously. That is to say, insect subclades present different rates of speciation and extinction. They found that certain biotic and abiotic factors did drive the diversification of the group and that diversity of feeding strategies impact directly in speciation and extinction rates of other guilds.

I found this study well thought-out, well-written and well executed. The methods are exhaustive and the authors tried to account for all the typical bias that can affect estimation of diversification rates. I particularly enjoyed the Limitation section included in the study. I recommend acceptance of this manuscript after they included some minor comments:

L105. You talk about peak of extinction rates but you report values of net diversification rate. I understand that the diversification rate is defined as speciation minus extinction rate, so diversification rates reflect the impact of extinction rates. But shouldn't it be better to report turnover rates or relative extinction or extinction rates if one wants to report the extinction intensity during the time periods studied?

It is confusing because you are using diversification rates to evidence the magnitude of the extinction events and to evidence the how quickly insect genera recover species richness (L141, 142.) Here I would definitely use turnover rates, which is a parameter that by definition express what you want to describe (Morlon, 2014; Ecology Letters)

L414-418. You mention that your first dataset was composed by 5,808 occurrences. Then you filter your dataset and eliminated synonyms, outdated combinations, nomina dubia, and other erroneous and doubtful records, and after correction you have a dataset of 17,250 occurrences. That is a dataset three times bigger. How is this possible? In addition, in lines 91 and 92 you say that your dataset was composed by 14,483 (family level) and 14,789 (genus level) occurrences. How these number match with the numbers described in methods?

Fig. 5B The shadow used in letters and in insects' silhouettes makes them look fuzzy

I am just curious about a trivial aspect. During the first five pages of the article, you use "speciation" rate to define the number of splitting events that give rise to species/Ma/lineage. From page 6 you use "origination" and I understand that you are referring also to define the number of splitting events that give rise to species/Ma/lineage. Do you use speciation and origination synonymously? It gives the impression that you are talking about different events, just please clarify.

Reviewer #1 (Remarks to the Author):

The paper by Jouault and colleagues entitled “Insect extinctions during the Permo-Triassic were driven by changes in floral assemblages and guilds’ interactions” represents a valuable contribution to the knowledge on the impact of the End-Permian Mass Extinction event on insects, and more generally on terrestrial invertebrates. In this paper, the authors performed extensive analyses relying on Bayesian inference exploiting the available insects fossil record and provide support for at least three events of extinctions between the Guadalupian and the Middle/Upper Triassic. The results in terms of origination/extinction rates of insect lineages inferred for the genus level almost reflect those obtained by Nicholson et al. 2015 for the family level using different datasets (doi:10.1371/journal.pone.0128554).

Thank you for reviewing and providing comments on our study. They have been very useful.

Yes, indeed the family-level pattern found in Nicholson et al. (2015) resembles what we found for the genus level. However, our pattern found at the family level strongly differs from their pattern (intensity and temporality of the shift). Also, it is difficult to compare the result of a pattern found for genus-level data with a pattern for family-level data (the data and the conclusion being strongly different). Furthermore, we bring more in-depth details on the heterogeneity of extinction and origination dynamics through time and their likely drivers. Nevertheless, we already cited this paper in our manuscript, but as you point it out, it is also relevant to cite the paper earlier in the manuscript.

The manuscript is well written, and the adopted methodologies are robust and appropriate. One of my minor notes concerns the fact that in the Results & Discussion there are several sentences providing information on the Methods, as well as some sentences that could be moved in the introduction. Thus, I suggest considering moving these sentences to the Methods section (e.g., lines 199-209) and Introduction section (e.g., lines 193-199). See also lines from 280 to 297 in the paragraph “Competition and facilitation within and between insect guilds”.

Thank you again for reviewing our manuscript and the positive opinion.

We agree with your concern for the lines indicated above. Their content was already present either in the *Methods* or in the *Introduction*, so we have now deleted them to avoid repetition or redundancy between the different parts of the manuscript.

The only part we prefer to maintain, although modified, is the “*Diversity dependence within and between insect guilds*”. We believe this section can be difficult to follow without a background and thus needs its own introductory information to facilitate readers’ interpretation. This choice also stems from the comments of reviewer #2.

Supplementary information, in some cases, should be improved in order to provide more detailed information (e.g. units of CO₂, O₂ concentrations). It seems that authors are not making publicly available the code for the performed analyses, it could be useful to allow repeating the adopted pipeline, but it will also help to understand the strength of the results.

Right, we have now provided the units of each variable in the Supplementary files, and added the code used for the analyses as a new Supplementary file.

Regarding the first two Results & Discussion sections, I have no concerns, while regarding the section “Ecological drivers of insect diversification and decline”, which in my view represents one of the two novelties presented in this manuscript, I have some doubts regarding the adopted methodology. First, the sentences in the Methods section “To test causal mechanisms of insect diversification ...” (line 596) is in my opinion misleading since to my knowledge the authors have inferred correlations, they are not legitimate to deduce cause-effect relationships. This aspect has to be considered even during the discussion of the achieved results.

Thank you for your comment. We agree with your concern and have rephrased the manuscript accordingly in the *Methods* section. Also, to avoid any over-interpretation, we have carefully reworded a few sentences in the core of the manuscript as follows:

- We changed “no study has investigated the causal mechanisms for the extinction and origination of insects” by “no study has investigated the correlations between the extinction and origination of insects”
- We changed “the MBD model indicated that several factors played a role in the diversification” to “the MBD model indicated that several factors are correlated with the diversification”

Furthermore, it is not clear if the authors have analysed the communities considering the different paleoenvironments in which they were established and if they have considered that these communities could differentially react to the same bioclimatic parameters.

Sorry for misleading your reading by using the word “communities”. We have now removed this word to avoid any misinterpretation.

We have analyzed taxonomic groups, and not paleoenvironments or local communities. We aim to provide a large-scale analysis (even if focused on two periods) of the diversification and extinction patterns of insects. Indeed, local responses to major paleoenvironmental changes may differ, but the fossil record of insects during the Permian and Triassic periods is, in our view, insufficient to draw any conclusions about local differences in impact. The major difference with the marine fossil record is the lack of deposits in the same area (spatial area such as a few kilometers) before and after the extinction event (Schachat and Labandeira, 2021: fig. 1). Insect Lagerstätten are known before and after extinction events, but in distant locations, hampering understanding of local changes. Similarly, in fossiliferous deposits of the Permian-Triassic, the banks of geological formations yield fossil insects only before or after the event or too small in quantities to derive any local differential impact. Hopefully, newly discovered deposits and future discoveries would allow conducting such fine-scale analyses at the Permian-Triassic boundary.

Note that this observation stands for the Permian and Triassic periods but not necessarily for the Jurassic or the Cretaceous. In fact, fine-scale studies of the changes in insect paleofauna are possible in certain parts of the world during the Cretaceous. For example, three amber deposits are known from the West Burma block (WBB), all different ages: Hkamti amber (early Albian ~110 Mya), Tanai amber (lowermost Cenomanian ~98 Mya), and Tilin amber (uppermost Campanian ~72.1 Ma). Therefore, it is possible to estimate the evolution of certain families present in these deposits (Jouault et al., 2022) and track the transition from the stem to crown groups of others (Perrichot, 2019).

Jouault, C., Maréchal, A., Condamine, F.L., Wang, B., Nel, A., Legendre, F., Perrichot, V., (2022) Including fossils in phylogeny: a glimpse into the evolution of the superfamily Evanioidea (Hymenoptera: Apocrita) under tip-dating and the fossilized birth–death process. *Zool. J. Linn. Soc.* 194, 1396–1423.

Perrichot, V., 2019. New Cretaceous records and the diversification of crown-group ants (Hymenoptera: Formicidae). 8th International Congress on Fossil Insects, Arthropods & AmberAt: Santo Domingo, Dominican Republic.

Schachat, S.R., Labandeira, C.C., 2021. Are insects heading toward their first mass extinction? Distinguishing turnover from crises in their fossil record. *Ann. Entomol. Soc. Am.* 114, 99–118.

In addition, the estimates of the bioclimatic parameters were obtained, I guess, from deposits which differ from those where the fossils were dug out. Is it thus possible to expand on a global

scale these local paleoclimatic data? Looking at the supplementary excel file - sheets atmospheric concentration of CO₂ and O₂, and Temperature (tables titles are not provided) – it is possible to note that in some cases, as for O₂, data are available with an interval of one million year. Could the author specify how they were estimated since it seems that values in the original papers have no regular intervals? Have the authors interpolated the values of two neighbouring measures? The same consideration could also be done for the temperature. In my view, the authors have to clarify the adopted strategy in the Methods section. In the case of CO₂ are present different values for the same million year (please specify the unit), which in some cases are quite different; how the author managed these measures? E.g. Age = 242.1 and CO₂ = 284, 244, 605. Are these values obtained from deposits characterized by different paleoenvironments? Have the authors considered these values as replicates? All the previously highlighted points make me doubtful about the strength of the achieved results and the inferred considerations.

Thank you for pointing out these issues. We reply to each point below:

- When revising our manuscript, we noticed that we indicated the wrong reference for the O₂ data. We have corrected the *Methods* section accordingly. We have also provided additional data and explanations for each variable.
- Yes, it is possible to expand these local paleoclimatic data on a global scale. For example, Westerhold et al. (2020) investigated the variation of the Cenozoic climate at a global scale. Using massive data from the International Ocean Drilling Program (<https://www.iodp.org/>) from different locations, it is possible to obtain a mean value of the global temperature or other variable like sea-level fluctuations (Miller et al. 2020).
- Sorry for the missing titles, they were deleted during the submission process by the submission website.
- We used the O₂ data of Lehtonen et al. (2017: Supplementary Figure 21) and not from Prokoph et al. (2008) as initially written. Note that this dataset is already publicly available at <https://github.com/dsilvestro/PyRate> (in the *example files* folder) and is reconstructed for 1-million-year time intervals.
- For the temperature, we used the data from Prokoph et al. (2008), converted to absolute temperatures following the methodology described in Condamine et al. (2019) (see section Global temperature variations through time in the latter reference). Condamine et al. (2019) constructed a dataset for more than 320 Mya but only provided the data for the Cretaceous and Cenozoic in their paper. Here, we used another part of their temperature curve, relevant for the timeframe of our study. These data reflect planetary-scale climatic trends, with time intervals inferior to 1-million-year, that can be expected to have led to temporally coordinated diversification changes in several clades rather than local or seasonal fluctuations.
- Our CO₂ data were obtained from Foster et al. (2017). They assembled an unprecedented dataset (for the last 420 million years) by cleaning 1,241 independent CO₂ estimates coming from five proxy and 112 published studies. We used their cleaned dataset and extracted all verified values for the Permo-Triassic interval. Because the initial data (i.e. independent estimates) were made in various locations for the same age, different values of the CO₂ concentration are provided. We incorporated all these values in our analysis, allowing PyRate to search for correlation for each value of the CO₂ concentration. We obtain a final correlation independent of the sampling location and fitting with our large-scale analysis. In any case, correlations with the CO₂ are extremely low as shown in the appended figure below (with a time interval plotted only for the period encompassing the three extinction events). We think that these low correlations stem from the scarce CO₂ sampling of the Permo-Triassic and we hope for future improvement of these data to corroborate or refine our results.

- Note that the correlations inferred by the MBD model are robust even when time intervals of the variables vary or are extremely short or heterogeneous (e.g., with numerous variations, see Lehtonen et al. 2017: fig. 4, Supplementary Figure 2).

Condamine F. L., Rolland J. & Morlon H. Assessing the causes of diversification slowdowns: Temperature-dependent and diversity-dependent models receive equivalent support. *Ecol. Lett.* 22, 1900–1912 (2019).

Foster, G., Royer, D. & Lunt, D. Future climate forcing potentially without precedent in the last 420 million years. *Nat. Commun.* 8, 14845 (2017).

Lehtonen S, et al. Environmentally driven extinction and opportunistic origination explain fern diversification patterns. *Sci. Rep.* 7, 4831 (2017).

Miller, K.G., et al. Cenozoic sea-level and cryospheric evolution from deep-sea geochemical and continental margin records 6, 1–15 (2020).

Prokoph, A., Shields, G. A. & Veizer, J. Compilation and time-series analysis of a marine carbonate $\delta^{18}\text{O}$, $\delta^{13}\text{C}$, $^{87}\text{Sr}/^{86}\text{Sr}$ and $\delta^{34}\text{S}$ database through Earth history. *Earth-Sci. Rev.* 87, 113–133 (2008).

Westerhold, T., et al. An astronomically dated record of Earth's climate and its predictability over the last 66 million years. *Science* 369, 1383–1387 (2020).

The paragraph “Limitations” in my view represents a highly appreciable addition, especially the one regarding the trophic guilds to which each taxon belongs.

Thank you. We believe that it is indeed a very important element for such analyses and have elaborated further regarding insect guilds (see also answers to reviewer #2 below).

Minor comments:

Lines 413-414. The fossil dataset used in the analyses includes occurrences spanning from Asselian to the Rhaetian while the bioclimatic parameters, used in the analyses, have a different time-span. Could you please explain this point?

Thank you for pointing out the heterogeneity. We initially provided our full compilation of data for the environmental variables (longer than those used in the analyses). For homogeneity purposes, we have now only provided the data for the period of interest, i.e. those used in the analyses.

The dataset retrieved by PBDB increased threefold after the addition of occurrences, non-included in PBDB, but published in the scientific literature. I suggest the addition of a supplementary table where the used literature is reported.

Thank you for noticing the three-fold increase. We have added a new sheet with all the references consulted and used for the present study.

Line 422. Please provide information on the unpublished taxa.

We have now clarified in the *Methods* section that all unpublished taxa are identifiable in our datasets by the "fam. nov." or the "gen. nov." noted after their names.

Lines 437-438. Please specify on which basis the author decided to attribute tentative species identifications characterized by "aff" or "?" to a higher taxonomic level. In the present version, authors reported "Tentative species identifications ... were most of the time included at a higher taxonomic level".

Sorry, we were unclear and did not correctly report what we meant. In the present study, we investigate the diversity dynamics of insects at the genus level and family level. Therefore, all the species with "aff" or "?" are already considered at the genus level and considered to be generic occurrences. We have clarified this point in the *Methods* section.

Lines 321-324. As an alternative, this pattern could be explained as an indirect effect of an increased amount of plant biomass or other not considered factors.

Indeed, our study proposes an entirely biological explanation of insect diversity variations through the Permo-Triassic interval, which relies on both abiotic and biotic factors, although our results may depend on our choice and availability of environmental and biological variables used as predictors. We anticipate that other proxies can provide alternative explanation to the observed pattern, and future studies with new data (like plant biomass through time) would be welcome to test such a hypothesis.

Reviewer #2 (Remarks to the Author):

The manuscript by Jouault et al. uses existing family- and genus-level records of fossil insects across the Asselian (lowermost Permian) to the Rhaetian (uppermost Triassic) periods from an existing database (<https://paleobiodb.org/>; PBDB) representing 3,636 species (1,784 genera, 418 and 418 families) and 17,250 total detections, after filtering. They use a suite of Bayesian estimation procedures using the process-modeling software, PyRate to estimate changes in diversity and diversification and extinction rates for genera and families for all insects and for each major ancestral clade. Finally, they cross-correlate changes in diversity (and rates therein) with various hypothesized drivers of change, including biotic, abiotic, vicariance events, and "ecological interactions."

The submitted manuscript is generally well-written and reports some interesting trends. I cannot evaluate all the specifics of modeling parameter choice, etc. as I am not specifically familiar with the PyRate program. However, the Methods were convincing and the stated choices appeared defensible.

Thank you for reviewing our manuscript and for the general positive opinion. Thank you also for pointing out your unfamiliarity with PyRate. In our response to your comments, we explain as much as possible how PyRate works and our motivation behind using this method for macroevolutionary analyses.

The Results were interesting and relevant overall. However, I found myself quite unconvinced with the framework and interpretation of "ecological interactions" as drivers of extinction/diversification, which was a major thrust of the paper (even appearing in the title). Ultimately, it is just too big of a stretch to interpret broad correlations among lineages belonging to coarse-scale feeding guilds as the author have done (neg. correlations as evidence of competition and positive as facilitation). For example, surely herbivores might respond negatively to major eruptions while detritivores (for a while, anyway) might benefit). However, this would be classed as competition in the current MS, which is quite unlikely (even impossible since the niches are distinct by definition) in this case. Unless the authors can make a much, much stronger case for the validity of these assumptions (and I don't think they can) I would suggest that work is unpublishable unless removed. This does remove a not-insignificant component of the manuscript but my feeling is that the remaining results have sufficient value to stand alone, though I am not 100% certain of their novelty.

Thank you for your positive feedback. Given your comment, we have first modified the title to better fit with the content of our article, following your comments and those of reviewer #1. This revised title does not highlight guild interactions anymore (see also the section *Limitations* in the main text in this regard). In addition, we have performed four additional analyses, which all concur with our previous results. Namely, one new MCDD analysis in which the *Generalists* are merged with *Detritivores/Fungivores* (these guilds are harder to delineate in the fossil record than predators and herbivores) to test whether inter-guild diversity dependence would also be recovered in a simpler ecological network. These new results show that herbivores hold a central position in the Permo–Triassic interaction network. Then, we cross-validated the MCDD results with three new MBD analyses as detailed in the revised manuscript and our responses below. Also, we are not convinced with your example: A major eruption would likely impact immediately all guilds and not necessarily only one or two of them. Nevertheless, we have completely rephrased this part of the manuscript to highlight the diversity dependence between and within guilds.

Overall, we take this general comment as an opportunity to explain the MCDD model (originally described in Silvestro et al. 2015). This model looks for diversity dependence within and between clades, it is not the reflection of a trophic network nor the quantification of the effect of an environmental variable (the MBD is used for that). We agree that some phrasing in the first version of our manuscript may have been misleading, and we have corrected them. The MCDD allows modelling how a clade “responds” to change of diversity in another clade. Under negative interactions (competition), increasing species diversity in one clade would decrease the speciation rates and/or increase the extinction rates in another. Each parameter expresses a diversity dependence relationship between the diversity of a clade and the speciation or extinction rates of another. Thus, the model infers directionality and magnitude of the reciprocal interactions between two clades.

Negative interactions under the MCDD model result in a reduction of a clade's speciation rate and/or an increase of its extinction rate. Mathematically, although a bit complicated to follow, it translates into $g^{\lambda}_{ij} > 0$ and $g^{\mu}_{ij} > 0$ indicating that the diversity of clade j (1) correlates negatively with the speciation rate of clade i, and (2) correlates positively with the extinction rate of clade i. On the contrary, $g^{\lambda}_{ij} < 0$ and $g^{\mu}_{ij} < 0$ indicate a positive interaction between clades, so that increasing diversity of a clade j correlates with higher speciation rates and lower extinction rates in clade i. Finally, $g^{\lambda}_{ij} = 0$ and $g^{\mu}_{ij} = 0$ imply that no diversity dependent effects are detected and the diversification dynamics of clade i is independent from the diversity of clade j.

More explicitly, the negative effect given by $g^{\lambda}_{ij} = 0.1$ and $g^{\mu}_{ij} = 0.2$ implies that the addition of one species in clade j will decrease the speciation rate in clade i by 10% of the

baseline rate (λ_i) and increase its extinction rate by 20% of the baseline rate (μ_i). Conversely, the extinction of one species in clade j will increase clade's i speciation rate and decrease its extinction rate by 10 and 20%, respectively.

Because of the uncertainty detailed in the *Limitations* section or in the *Multiple clade diversity-dependence model* section of the *Methods* section, we do not discuss the estimated values of the increase/decrease of extinction or origination rates *per se* but rather highlight the main trends inferred by the MCDD model. Coming back to your previous example “*herbivores might respond negatively to major eruptions (...) be classed as competition*”, the MCDD will not correlate the extinction of the herbivores with environmental changes but rather will infer the effect of the diversity decrease in herbivores with the evolutionary processes of other guilds. It can thus help address questions such as: Does decreased diversity in herbivores correlate with decreased origination rates in predators and increased extinction rates? The MCDD model can provide estimates to assess this question: if $g_{ij}^{\lambda} > 0$, the decreased diversity of herbivores induces higher origination rates in predators (considered as a negative interaction), but if $g_{ij}^{\lambda} < 0$, the decreased diversity of herbivores induces lower origination rates in predators (considered as a positive interaction). In this example, g_{ij}^{λ} is the correlation parameter of origination of predators (clade i) as impacted by the diversity of herbivores (clade j).

We took advantage of the revision process to strengthen the results of this part. Because the interactions we found in our first analysis were focused on the herbivores, we have now simplified our model to three guilds by merging *Generalists* with *Detritivores/Fungivores* and conducting additional tests:

1) We performed another MCDD analysis with three guilds (and not four) to confirm that the interactions are still centered around herbivores even when the number of guilds is reduced. This analysis was also conducted because it strengthens the delineation of the different guilds (delineating *Generalists* and *Detritivores/Fungivores* is challenging).

2) We performed three additional analyses with the MBD model using the diversity of the different guilds as explanatory variables. These analyses allow the investigation of diversity dependence between the three guilds using a different birth-death model.

While the MS is well-written and structured, it is hard to get through and confusing in places. Some things that might help include:

- A figure or table with all the date estimates (which is available not until L656 in the Methods). This would make keeping track of the various epochs and transitions much easier for the general reader.

We understand that keeping track of ages might be challenging for readers unfamiliar with them and this is exactly why we gave the most important ones in the *Abstract*, and in the *Introduction*. Most of the time they are indicated next to the event we defined or mentioned:

- Line 40: The most dramatic of these extinctions occurs at the boundary (...) ca. 252 million years ago (Ma) (...)

- Line 44: (...) the Guadalupian extinction event (GEE), which occurred ca. 260.5 Ma; and the Carnian pluvial episode (CPE) that occurred between 234 to 232 Ma (...)

- Line 77: Asselian to Rhaetian timespan (i.e. between 298.9 and 201.3 Ma)

We hope that, combined with ages given in the *Methods* and depicted in the different figures, this presentation would be clearer and straightforward for readers.

- An Info box or glossary to outline the different insect taxonomic groupings you are using (and perhaps a short explanation as to why for each).

We understand this comment. However, the format of the journal does not allow using boxes as seen in some other journals. Please note, however, that guilds are explained in the *Methods* section (“We estimated the past diversity dynamics for three (...) distinguished herbivorous and

carnivorous taxa among Hemiptera") and provided in Supplementary Files 2. We have also better explained how 'predators' and 'herbivores' guilds were delineated.

•Explanatory text to accompany the icons, especially in Fig. 3.

We agree with your comment, and we have added a legend for all the different pictograms used in this figure.

I'm confused by Fig. 1 -- what is driving rapid and non-trivial changes in lineage richness (1F) when no such changes are apparent in D and E? D and E are model outputs I gather, but this discrepancy makes me wonder if the lack of corresponding ups and downs reflects a sample size artifact rather than the lack of a pattern that roughly conforms what is seen for genera. Suggest clarifying for readers.

We are sorry for the confusion. Here we try to explain the results depicted on Figure 1. Figure 1D shows the origination (blue) and extinction (red) rates, and Figure 1E shows the net diversification rate (origination minus extinction) inferred from the fossil occurrences. Figure 1F represents the number of lineages through time (LTT). Based on fossil occurrences and the preservation process, PyRate estimates the ages of taxon appearance and disappearance for all taxa in the dataset. These ages determine the lifespan of taxa, and then allow the computation of the diversification rates (as origination and extinction rates, expressed as events per Myr) and the LTT. The lack of corresponding ups and downs between rates and LTT simply comes from two facts: (1) the RJMCMC model likely misses small shifts of origination and/or extinction to explain the fine-scale ups and downs in taxonomic diversity (there is a smoothing of the rates), and (2) there is a lag between diversification rates and changes in diversity, as it takes time for rates to affect diversity (protracted effect). We have clarified these points in the manuscript.

Note that sample sizes in genus- and family-level analyses are similar (14,789 occurrences at the genus level and 14,483 at the family level), and the general pattern of the LTT plots are also similar. The main difference between the two is the longevity (or the life span) of families, which is conspicuously longer than that of genera. Therefore, our results shows that the families of insects that appeared during the Permian survived the P/T event and diet out later during the Triassic, even if their presences are not always directly evidenced by direct occurrences.

Also, what is the correct scale from the perspective of both time and magnitude for consideration of impacts on diversity? For example, does the definition of a major extinction include how much net diversification rate dips below zero and for how long. How far below zero does it need to go, and for how long? Does a subsequent recovery influence the interpretation of such an event, or for that matter, might a downturn in diversification following a period of elevated diversification (as appears to be the case for the P/T and L/C event) be better considering a correction? Given the magnitudes of these two events, I'm not really suggesting this latter point as a viable explanation, but what is the interpretation of the spikes before the drops?

Thank you for this comment. Currently, the definition of what is a mass extinction dates to Sepkoski (1986), stating that a mass extinction is 75% of diversity loss, is short in time (geologically speaking), is geographically widespread, and affects most clades. However, such a 'definition' does not include how much net diversification rate must dip below zero. As you can see from the different figures throughout the manuscript, there are background origination and extinction rates (i.e. in 'normal' conditions). This means that genera died out and diversified continuously. A mass extinction is witnessed as a sudden change in the extinction rate clearly exceeding the background rate. Additionally, all the events we study here are already

considered to be periods of major biodiversity loss, and at least one is a mass extinction (at 252 Mya). The recovery does not influence the interpretation of such an event, and the high diversification rates recorded after the P/T or the L/C cannot be interpreted as a correction, but as a re-diversification period following a drastic loss.

In fact, the faunas (taxonomic composition, morphology of the different taxa) are very different. Ongoing studies of the wing shape and the wing venation of Odonoptera –a group of insects whose morphology greatly changes during the Permo-Triassic– shows that the morphology of Permian fauna greatly differs from that of Triassic fauna, with different morphospaces. A similar situation is observed for the Coleoptera, for which Zhao et al. (2021) showed that both the taxonomic diversity and morphological disparity dropped dramatically during the Early Triassic: all xylophagous stem-group beetles become extinct near the Permian-Triassic boundary or abruptly decreased in the Early Triassic, while aquatic phoroschizid and ademosynid lineages crossed the Permian/Triassic boundary and diversified in the Middle Triassic. Coleoptera recovered their taxonomic diversity during the Middle Triassic by the rise of new predatory and herbivorous groups –absent from the Permian period– and synchronized with the recovery of terrestrial ecosystems. Therefore, these events correspond with the extinction of entire fauna and the diversification of new ones. We explain this transition in our manuscript with the extinction of entire orders (such as the Paleodictyoptera) and the diversification of new ones (such as the Hemiptera). The simultaneous increase of extinction and origination rates at the LPME can be interpreted as a turnover of fauna (*sensu* Schachat and Labandeira 2021) and the peak of origination at the end of the Anisian (Fig. 1A) as the diversification of a new fauna in new ecosystems (recovery). We have clarified these points in the manuscript.

Schachat, S.R., Labandeira, C.C., 2021. Are insects heading toward their first mass extinction? Distinguishing turnover from crises in their fossil record. *Ann. Entomol. Soc. Am.* 114, 99–118.

Zhao, X., et al. Early evolution of beetles regulated by the end-Permian deforestation. *eLife* 10, e72692 (2021).

Specific comments:

Throughout MS: Diversity is a specific term in ecology that generally combines the concepts of species richness and their relative abundances (evenness). I think most of the many places the word "diversity" is used, "richness" might be better. From my perspective, it would be best to make this change throughout. However, I recognize that terminology usage can vary across sub-disciplines, but please consider.

Thank you for your comment. Indeed, the terminology depends strongly on the discipline. The present paper deals with macroevolution and we use the term "diversity" as taxonomic diversity (e.g. genus richness). For the sake of consistency and to respect the terminology of the discipline, we prefer to keep "diversity" and we use "taxonomic diversity" in the *Introduction* for its first usage to make things clearer.

L79: "genera and at the stage- or formation-level for taxon ages." Something is wrong with this sentence.

Corrected.

L84: "co-occurring guilds" Remove "co-occurring"

We have modified the sentence.

L98: "diversity dynamic" Prefer "insect generic richness", "diversification rate", or other term

-- "diversity dynamic" doesn't work in this context. Diversity and/or rates can go up or down, but a dynamic just that -- it can't decline as you've stated here. I see that this usage is consistent throughout the MS - I would strongly recommend changing (but will not comment at every usage).

As with the term “diversity”, the term “diversity dynamic” is widely used in macroevolution (e.g., Quental and Marshall, 2010; Romano et al. 2014; Gorzelak et al., 2015; Flannery-Sutherland et al., 2022). As explained above, we keep our formulation throughout the entire manuscript for reasons of homogeneity in the discipline given that there is now a definition.

Flannery-Sutherland, J. T., Silvestro, D. & Benton, M. J. Global diversity dynamics in the fossil record are regionally heterogeneous. *Nat. Commun.* 13, 2751 (2022).

Gorzelak, P., et al., 2015. Diversity dynamics of post-Palaeozoic crinoids – in quest of the factors affecting crinoid macroevolution. *Lethaia* 49, 231-244.

Quental, T.D., Marshall, C.R., 2010. Diversity dynamics: molecular phylogenies need the fossil record. *Trends Ecol. Evol.* 25, 434-441.

Romano, C., et al. Permian–Triassic Osteichthyes (bony fishes): diversity dynamics and body size evolution. *Biol. Rev.* 91, 106–147 (2014).

L84: "We simultaneously assessed the effect of co-occurring guilds (herbivores, predators, detritivores/ fungivores, and generalists) on their speciation and extinction rates by quantitatively investigating the roles of competition among insect clades throughout the Permian and Triassic periods. " While I think it's very interesting to consider how major extinction events have manifested similar v. uniquely patterns across these broad feeding guild categories, I am again very dubious that using such a coarse filter can tell you much about positive or negative interactions, or their importance in speciation and extinction dynamics.

Thank you for your comment. We have completely revised this part of the manuscript. We now emphasize diversity dependence within and between the guilds. The revised text highlights the central position that herbivores hold in the Permo–Triassic interaction network. Now, we only suggest the competition or the facilitation as results analogous to the interactions found in extant ecosystems.

L106 (and throughout): "ca." I think you should state these rates without saying "ca." each time. It is implicit that these are estimates. Providing some estimates of uncertainty (i.e., 95% CI range) would be useful though.

As you know, *ca.* is the abbreviation of the Latin word *circa* meaning “close to” or “around”. In our manuscript we use *ca.* to provide an approximate value of the exact value (example: 291.235 Ma ==> *ca.* 291.2 Ma). To avoid any confusion, we have replaced the *ca.* by \approx .

L106: "events/ Ma/ lineage" I assume that you mean "events/million years/lineage" in which case "Ma" is not correct.

The abbreviation “Ma” is commonly used in the literature dealing with evolution, including in Nature journals (and many others), which means “million years ago” or “Mega Annum”. In geology a debate remains open concerning the use of Myr (duration) *plus* Ma (million years ago) *versus* the use of the term Ma only. In either case the term Ma is used in geology literature conforming to ISO 31-1 (now ISO 80000-3) and NIST 811 recommended practices.

L181: " However, during the LPME, age had a strong effect on extinction ($\phi = 9.1677$, 95% [CI] = 2.3886-18.7343; Supplementary Table S2). " I actually find this compelling but you said in the paragraph above that model convergence is difficult to reach during unstable periods. Could this finding be an artifact of violating the assumptions of stability?

Thank you for your comment. Indeed, we underlined that convergence is difficult to reach in such situations. It is a reality and we wanted scholars, who might want to repeat the analyses, to be aware of this (and we wished to be transparent to readers). However, convergence can be achieved (which is the case in our analysis) by multiplying the number of replicates of the analyses and checking the parameters for convergence. To clearly separate the *Methods* (and related comments) from the main text, we moved this section to the *Methods*.

L196: "Insects' past dynamic is inevitably linked to environmental changes, which directly led to their diversification or extinction" This statement is too strong. Sure, environments matter, but vicariance events and novel associations with host plants could be at least as strong a driver in the case of diversification, as is borne out in Fig. 3.

Right, we have tempered our point and added more flexibility to our sentence.

L208: "and four abiotic variables (global temperature variations; global variation of atmospheric CO₂ and O₂; and continental fragmentation)" I assume that you mean temporal (and not spatial) variation (not plural), but if so, at what temporal scale did you calculate? This matters. It also might be appropriate to test for lag effects.

For each variable, we provided the data for the temporal variation of the studied variables as the input values of the MBD model. Unfortunately, we cannot consider the spatial variations of the environmental variables in PyRate because models and biome reconstructions are currently not available for all the periods or for the entire surface of Earth. However, we do agree that spatial heterogeneity of a given variable does matter. In addition, the lag effect of an environmental variable in the deep past ecosystems would be difficult to investigate compared to studies in current ecosystems. When ecological studies investigate lag effect in extant ecosystems, they often investigate this effect for months or years while the time interval of our variables is about 1 million years.

L227: "fluctuation in the diversity of non-Polypodiales would have accelerated their extinction (Fig. 3A, B)." Not "fluctuation" but "increase", no? But I'm confused, aren't these diversity categories overlapping (i.e., Gymnosperm diversity is a subset of non-Polypodiales diversity). It seems more likely that this refers to non-Polypodiales ferns, but this is not clear.

Yes, you are completely right, we referred to the non-Polypodiales ferns. We have corrected the manuscript, accordingly, thank you. This is not really a drastic increase (see Lehtonen et al. 2017) but rather a fluctuation. Therefore, we used this term only for this sentence.

L260: "The amber production of the Carnian might also reflect the beginning of resin production as a defense strategy against phytophagous insects, maybe following the rise of modern insects⁶⁶. " Does this bias the database toward detection of gymnosperm-associated taxa?

No, because to date there is no amber outcrop with insect fossil inclusion formally described before the Cretaceous (maybe one Jurassic amber deposit with insect is known but to date not published and the data is not certain). The inclusions embedded in Triassic ambers mainly represent mites or other Acariformes assumed to feed on plants (Schmidt et al. 2012; Sidorchuk et al. 2015).

Schmidt, A. R., et al. Arthropods in amber from the Triassic Period. *PNAS*. 109, 14796–14801 (2012).

Sidorchuk, E. A., et al. Plant-feeding mite diversity in Triassic amber (Acari: Tetranychidae). *J. Syst. Palaeontol.* 13, 129–151 (2015).

L288: "Similarly, detritivores/ fungivores and generalists may compete with herbivores" What is a generalist insect in this context? Usually, they are herbivorous but with fewer restrictions on host plant breadth. This needs to be clarified. Either way, I think it's a major stretch to hypothesize competition among highly distinct feeding guilds. By what mechanism?

Thank you for your comment. You are right, in the first version mentioning inter-guild competition was inaccurate. We acknowledge the lack of clarity on this aspect in the previous version and we, therefore, corrected the situation in the revised manuscript. We also fused the two guilds with relatively poor delineations (*Generalists* and *Detritivores/Fungivores*) to perform a new MCDD analysis. Also, we used the term generalist to refer at taxa with opportunistic habits and able to feed on a variety of food sources, like the Dermaptera that feed on plants but also on small insects or soil debris (Crumb et al. 1941). We do not investigate the generalist (fewer restrictions on the host plant) vs. specialist (dependent on a single plant) behavior within each guild.

Crumb, S. E., Eide, P. M. & Bonn, A. E. The European earwig. U.S Department of Agriculture (1941).

L303: ", suggesting intra-clade competition" Big leap to get to intra-clade competition, which I don't really buy. The pattern is interesting though. From L303-348 is where the MS really lost me in terms of its (indefensible, in my opinion) interpretation of cross-correlations as evidence of ecological interaction. I would strongly advocate for deletion or major reworking of these paragraphs.

Thank you for your comment. We have modified this part of the manuscript to emphasize the diversity dependence rather than extrapolating on the competition or other interactions. We only proposed competition as a possible explanation for our results—i.e. not the only possible explanation—and we restricted competition within guilds (intra-guild diversity dependence).

L441: "ichnospecies" Define

We provide an example of ichnospecies and add a definition.

L444: "taxonomic 'bins'." I would explain what you mean here.

We "replaced taxonomic 'bins'" with "wastebaskets" and added a detailed explanation for the example used to illustrate "wastebaskets" (viz. Grylloblattodea) by pointing out the limits of these groups and the problem of their delineation and non-monophyly.

L494: "(1) the parameters of the preservation process" Explain.

Representing the expected number of occurrences per lineage per million years, preservation (denoted as q in PyRate) is modelled on a lineage-specific basis and through time (Silvestro et al. 2014). PyRate incorporates heterogeneity in preservation rates across lineages (Gamma model) and a non-homogeneous Poisson process (NHPP) in which preservation rate changes during the life span of each lineage, following a bell-shaped trajectory as estimated from the data (Silvestro et al. 2014). In the latest version of PyRate (Silvestro et al. 2019), preservation can be now modelled with a time-variable Poisson process (TPP), in which preservation rates vary across time windows (meaning that there are as many q as time bins defined by the users). Hence, PyRate can have more than two parameters for the preservation process. In the *Methods* section (*Dynamics of origination and extinction*), an explanation and definition of the preservation process can be found ("All analyses were set with the best-fit preservation process (...) appropriate when rates over time are heterogeneous").

Silvestro, D., et al. PyRate: a new program to estimate speciation and extinction rates from incomplete fossil data. *Methods Ecol. Evol.* 5, 1126-1131 (2014).

Silvestro, D., et al. Improved estimation of macroevolutionary rates from fossil data using a Bayesian framework. *Paleobiology* 45, 546-570 (2019).

L506: "Therefore" Is the "Therefore," needed?

Not really, deleted.

L624: "absolute temperatures " Wouldn't variation or change in T be more useful?

It corresponds to variation of temperatures (denoted as T) at a global scale inferred from $\Delta^{18}\text{O}$ data of Prokoph et al. (2008). These data were converted to absolute temperatures following the methodology described in Condamine et al. (2019) (see section “*Global temperature variations through time*” in the latter reference). These data reflect planetary-scale climatic trends, with time intervals inferior to 1-million-years, that can be expected to have led to temporally coordinated diversification changes in several clades rather than local or seasonal fluctuations.

Reviewer #3 (Remarks to the Author):

Dear Authors:

This paper represents a great and valuable contribution to diversity dynamics of insects during Permian–Triassic times. The manuscript investigates and describes several important biotic and abiotic factors for explain the diversity of insects during Permian and Triassic. Also, this contribution is very important because never was investigated the diversity decline of insects to genus level, as you commented in the manuscript.

This paper merits publication because builds on previous research and well-developed methodologies, the objectives are clear, the methods used were appropriate, sound, and employed correctly. The multiples statistical test used are concordant with the results, the authors interpret general pattern with caution.

The authors made an exhaustive revision of available information on fossil insects.

The paper is generally well-organized and well-written (the authors use grammar and syntax correctly), but please, bear in mind I am not a native English speaker.

The references are adequate, current and pertinent. All illustrations are necessary and are referred to in the body of the text.

To summarize this is a very interesting paper in all regards.

Thank you for reviewing our manuscript and for your positive feedback.

Some minor corrections/suggestions:

-Line 42: replace colloquially called by commonly named.

Corrected.

-Line 46: after warming add a comma (,)

Added.

-Line 48: after e.g. add a comma (e.g.,). Please check in all text.

Added.

-Line 102: replace (Fig. 1–F) by (Fig. 1D–F).

Corrected.

-Line 136: replace old orders by ancient orders.

Corrected.

-Line 136: after i.e. add a comma (i.e.,). Please check in all text.

Added.

-Line 160, Line 695: replace well known by well-known.

Corrected.

-Line 160: replace and the pace of species description by but the description of the species is comparatively slow...

We have deleted the sentence for clarity.

-Line 161 replace Cretaceous or the Cenozoic descriptions by Cretaceous and Cenozoic ones.

Corrected.

-Line 201: replace Permian-Triassic by Permian–Triassic. Please check in text the em-dash.

Corrected, also corrected for Roadian–Wordian and Ladinian–Carnian.

-Line 217: replace diversities by diversity.

Corrected.

-Line 217-218: replace Polypodiales have significantly affected insect diversification by ...and non-Polypodiales significantly affected to insect diversification...

Corrected.

-Line 253: replace concurs with by agree with or coincide with.

Corrected.

-Line 458: replace or of an insufficient by or insufficient number of...

Corrected.

-Line 464: replace signal by evidence

Corrected.

-Line 562: delete the in (the Roadian-Wordian, the LPME, and the Ladinian-Carnian) and replace - by –

Corrected.

-Line 627: replace well recorded by well-recorded

Corrected.

-Line 677-638: replace time through correlations with environmental variables by in relation to environmental variables

Corrected.

-Line 688: replace buccal pieces by mouthparts

Corrected.

-Line 718: add to after thank

Corrected.

-Line 722: add comma after A.N and C.J.

Corrected.

I hope the authors work on the few suggestions and get this paper published soon because it is excellent contribution to Palaeoentomology. Let me know if there is anything else I can help with.

Best regards.

Thank you again. We have modified our manuscript according to all your corrections/suggestions. We also hope that this study will contribute to better understand the dynamics of insects in the deep time.

Reviewer #4 (Remarks to the Author):

Comments to the authors

This study estimates if the magnitude that well-known extinction events had in insect's diversification. Also, they test if a number of biotic and abiotic factors promoted speciation or extinction rates within the different subclades of Insecta. Finally, they evaluate if feeding strategies are correlated with speciation/extinction rates and how the diversity of each strategy affects the rates of the other strategies.

As a result, they found that the well-known mass extinction events affected insect's diversification heterogeneously. That is to say, insect subclades present different rates of speciation and extinction. They found that certain biotic and abiotic factors did drive the diversification of the group and that diversity of feeding strategies impact directly in speciation and extinction rates of other guilds.

I found this study well thought-out, well-written and well executed. The methods are exhaustive and the authors tried to account for all the typical bias that can affect estimation of diversification rates. I particularly enjoyed the Limitation section included in the study. I recommend acceptance of this manuscript after they included some minor comments:

We are grateful for the positive feedback, your enthusiasm, and appreciate your comments and suggestions that contribute to improve our manuscript.

L105. You talk about peak of extinction rates but you report values of net diversification rate. I understand that the diversification rate is defined as speciation minus extinction rate, so diversification rates reflect the impact of extinction rates. But shouldn't it be better to report turnover rates or relative extinction or extinction rates if one wants to report the extinction intensity during the time periods studied?

It is confusing because you are using diversification rates to evidence the magnitude of the extinction events and to evidence the how quickly insect genera recover species richness (L141, 142.) Here I would definitely use turnover rates, which is a parameter that by definition express what you want to describe (Morlon, 2014; Ecology Letters)

Thank you for your comment. We acknowledge the non-intuitive formulation in some cases. Therefore, we chose to remove net diversification rate values, when misleading, and rather used a magnitude comparison (how many times the extinction is higher than the background rate). The background rate is the rate in "normal conditions" viz. not during the extinction or diversification peaks. For example, at the genus level, the extinction has a background rate of 0.1318 events/Ma/lineage for the period encompassing the three extinction events and, during the LPME, it increases to 0.5677 events/Ma/lineage. Thus, the extinction rate at the LPME is 4.3-fold higher than the one in "normal conditions", suggesting a major extinction event. We believe that such formulations will be more meaningful to readers and better highlight the extinction events.

L414-418. You mention that your first dataset was composed by 5,808 occurrences. Then you filter your dataset and eliminated synonyms, outdated combinations, nomina dubia, and other erroneous and doubtful records, and after correction you have a dataset of 17,250 occurrences. That is a dataset three times bigger. How is this possible? In addition, in lines 91 and 92 you say that your dataset was composed by 14,483 (family level) and 14,789 (genus level) occurrences. How these number match with the numbers described in methods?

The difference recorded between the initial dataset and our final dataset stems from PBDB (<https://paleobiodb.org>) itself. Most of the time, occurrences in PBDB (for fossil insects) reflect the number of localities/deposits in which a given species is known and not the number of known specimens per species. This results in a huge discrepancy between the known number of specimens and the occurrences in PBDB. For example, the species *Voltziaephemera fossoria* Sinitshenkova and Marchal-Papier 2005 is known by four occurrences in PBDB (corresponding to the four localities of the species: Bust, Adamswiller, Arzviller, and Vilsberg), while 250 specimens are known to date (Sinitshenkova et al., 2005). Ten type specimens *plus* 240 other specimens are documented in the original description of *V. fossoria* (Sinitshenkova et al., 2005, p. 384). In other words, the occurrences in PBDB do not consider either type series or additional material. This example is not an isolated case in the insect fossil record as also exemplified, for instance, in Plecoptera (Jouault et al., 2022). The species *Plutopteryx beata* Sinitshenkova, 1985 is known from more than 1,400 specimens but only one occurrence is present in PBDB (Sinichenkova, 1985, p. 133).

Jouault, C., Nel, A., Legendre, F. & Condamine, F. L. 2022. Estimating the drivers of diversification of stoneflies through time and the limits of their fossil record. *Insect Syst. Div.* 6: 1–14.

Sinichenkova, N.D., 1985. New Jurassic Stone Flies of the family Baleyopterygidae. *Paleontol. J.* 18:129–134.

Sinitshenkova, N.D., Marchal-Papier, F., Grauvogel-Stamm, L., Gall, J.C., 2005. The Ephemeroidea (Insecta) from the Gres a Voltzia (early Middle Triassic) of the Vosges (NE France). *Paläontologische Zeitschrift* 79:377–397.

The number 17,250 corresponds to the occurrences of Insecta gathered for the Permo-Triassic. This number encompasses numerous specimens (e.g., fragmentary wings, legs) that cannot be attributed to a particular family of genera. Therefore, they are part of our dataset but not included in our analyses. This explains the difference between the number 17,250 and the number 14,789, which represents these dubious specimens. Because they are indicative of the presence of Insecta and may be used for future studies we mentioned their presence in the manuscript. We hope that this clarification helps to better understand the value of our datasets.

Fig. 5B The shadow used in letters and in insects' silhouettes makes them look fuzzy
Yes, we agree and have removed the shadows.

I am just curious about a trivial aspect. During the first five pages of the article, you use “speciation” rate to define the number of splitting events that give rise to species/Ma/lineage. From page 6 you use “origination” and I understand that you are referring also to define the number of splitting events that give rise to species/Ma/lineage. Do you use speciation and origination synonymously? It gives the impression that you are talking about different events, just please clarify.

Thank you for pointing out the lack of definition for “origination”. We consider the origination as the speciation at the genus level (pace of genus appearance through time). We have now included this definition in the main text.

REVIEWERS' COMMENTS

Reviewer #1 (Remarks to the Author):

The authors properly clarified all my previous concerns and, in the present form, the manuscript is improved. I have no further suggestions for this manuscript.

MM

Reviewer #2 (Remarks to the Author):

The manuscript by Condamine et al. uses existing family- and genus-level records of fossil insects across the Asselian (lowermost Permian) to the Rhaetian (uppermost Triassic) periods from an existing database (<https://paleobiodb.org/>; PBDB) representing 3,636 species (1,784 genera, 418 and 418 families) and 17,250 total detections, after filtering. They use a suite of Bayesian estimation procedures using the process-modeling software, PyRate to estimate changes in diversity and diversification and extinction rates for genera and families for all insects and for each major ancestral clade. Finally, they cross-correlate changes in diversity (and rates therein) with various hypothesized drivers of change, including biotic, abiotic, vicariance events, and interguild correlations in rates.

It is clear from some of their responses that our sub-fields are discrete enough that preferred terminology varies somewhat, but I defer to the authors' choices. I appreciate the conceptual shift in the interpretation rate correlations, I think it would be good to take 1-2 more sentences to fully explain the new term ("diversity dependence") at L67 as they have done in the response to reviewers document.

This is a resubmission of a manuscript I had reviewed earlier. I am duly impressed with the thoroughness and thoughtfulness of the response and corrections. The major issues have been mitigated, and I think the manuscript would make a solid contribution to the literature.

Reviewer #3 (Remarks to the Author):

-No further comments submitted-

Reviewer #4 (Remarks to the Author):

Authors have clarified my doubts and followed my suggestions. I recommend acceptance of the manuscript.

Reviewer #1 (Remarks to the Author):

The authors properly clarified all my previous concerns and, in the present form, the manuscript is improved. I have no further suggestions for this manuscript.

MM

Thank you again for reviewing our manuscript and for your positive opinion.

Reviewer #2 (Remarks to the Author):

The manuscript by Condamine et al. uses existing family- and genus-level records of fossil insects across the Asselian (lowermost Permian) to the Rhaetian (uppermost Triassic) periods from an existing database (<https://paleobiodb.org/>; PBDB) representing 3,636 species (1,784 genera, 418 and 418 families) and 17,250 total detections, after filtering. They use a suite of Bayesian estimation procedures using the process-modeling software, PyRate to estimate changes in diversity and diversification and extinction rates for genera and families for all insects and for each major ancestral clade. Finally, they cross-correlate changes in diversity (and rates therein) with various hypothesized drivers of change, including biotic, abiotic, vicariance events, and interguild correlations in rates.

It is clear from some of their responses that our sub-fields are discrete enough that preferred terminology varies somewhat, but I defer to the authors' choices. I appreciate the conceptual shift in the interpretation rate correlations, I think it would be good to take 1-2 more sentences to fully explain the new term ("diversity dependence") at L67 as they have done in the response to reviewers document.

This is a resubmission of a manuscript I had reviewed earlier. I am duly impressed with the thoroughness and thoughtfulness of the response and corrections. The major issues have been mitigated, and I think the manuscript would make a solid contribution to the literature.

Thank you for reviewing our manuscript a second time and for your positive opinion.

We agree with your comments. We have now added a definition of the "diversity dependence" when it is first mentioned in the introduction.

Reviewer #3 (Remarks to the Author):

-No further comments submitted-

Thank you for approving our manuscript.

Reviewer #4 (Remarks to the Author):

Authors have clarified my doubts and followed my suggestions. I recommend acceptance of the manuscript.

Thank you for your second review and for your positive opinion.